# Local plasticity rules can learn deep representations using self-supervised contrastive predictions

**Bernd Illing**              **Jean Ventura**

**Guillaume Bellec***              **Wulfram Gerstner***

`{firstname.lastname}@epfl.ch`

Department of Computer Science & Department of Life Sciences
École Polytechnique Fédérale de Lausanne
1015 Switzerland

## Abstract

Learning in the brain is poorly understood and learning rules that respect biological constraints, yet yield deep hierarchical representations, are still unknown. Here, we propose a learning rule that takes inspiration from neuroscience and recent advances in self-supervised deep learning. Learning minimizes a simple layer-specific loss function and does not need to back-propagate error signals within or between layers. Instead, weight updates follow a local, Hebbian, learning rule that only depends on pre- and post-synaptic neuronal activity, predictive dendritic input and widely broadcasted modulation factors which are identical for large groups of neurons. The learning rule applies contrastive predictive learning to a causal, biological setting using saccades (i.e. rapid shifts in gaze direction). We find that networks trained with this self-supervised and local rule build deep hierarchical representations of images, speech and video.

## 1 Introduction

Synaptic connection strengths in the brain are thought to change according to 'Hebbian' plasticity rules [Hebb, 1949]. Such rules are local and depend only on the recent state of the pre- and post-synaptic neurons [Sjöström et al., 2001, Caporale and Dan, 2008, Markram et al., 2011], potentially modulated by a third factor related to reward, attention or other high-level signals [Kuśmierz et al., 2017, Gerstner et al., 2018]. Therefore, one appealing hypothesis is that *representation learning in sensory cortices emerges from local and unsupervised plasticity rules*.

Following a common definition in the field [Fukushima, 1988, Riesenhuber and Poggio, 1999, LeCun, 2012, Lillicrap et al., 2020], a hierarchical representation (i) builds higher-level features out of lower-level ones, and (ii) provides more useful features in higher layers. Now there seems to be a substantial gap between the rich hierarchical representations observed in the cortex and the representations emerging from local plasticity rules implementing principal/independent component analysis [Oja, 1982, Hyvärinen and Oja, 1998], sparse coding [Olshausen and Field, 1997, Rozell et al., 2008] or slow-feature analysis [Földiák, 1991, Wiskott and Sejnowski, 2002, Sprekeler et al., 2007]. Hebbian rules seem to struggle especially when 'stacked', i.e. when asked to learn deep, hierarchical representations.

---

*shared last author

This performance gap is puzzling because there are learning rules, relying on back-propagation (BP), that *can* build hierarchical representations similar to those found in visual cortex [Yamins et al., 2014, Zhuang et al., 2021]. Although some progress towards biologically plausible implementations of back-propagation has been made [Lillicrap et al., 2016, Guerguiev et al., 2017, Sacramento et al., 2018, Payeur et al., 2021], most models rely either on a neuron-specific error signal that needs to be transmitted by a separate error network [Crick, 1989, Amit, 2019, Kunin et al., 2020], or time-multiplexing feedforward and error signals [Lillicrap et al., 2020, Payeur et al., 2021]. Algorithms like contrastive divergence [Hinton, 2002], contrastive Hebbian learning [Xie and Seung, 2003] or equilibrium propagation [Scellier and Bengio, 2017] use local activity exclusively to calculate updates, but they require to wait for convergence to an equilibrium which is not appropriate for online learning from quickly varying inputs.

The present paper demonstrates that deep representations can emerge from a local, biologically plausible and unsupervised learning rule, by integrating two important insights from neuroscience: First, we focus on self-supervised learning from temporal data – as opposed to supervised learning from labelled examples – because this comes closest to natural data, perceived by real biological agents, and because the temporal structure of natural stimuli is a rich source of information. In particular, we exploit the self-awareness of typical, self-generated changes of gaze direction ('*saccades*') to distinguish input from a moving object during fixation from input arriving after a saccade towards a new object. In our plasticity rule, a global factor modulates plasticity, depending on the presence or absence of such saccades. Although we do not model the precise circuit that computes this global factor, we see it related to global, saccade-specific signals from motor areas, combined with surprise or prediction error, as in other models of synaptic plasticity [Angela and Dayan, 2005, Nassar et al., 2012, Heilbron and Meyniel, 2019, Liakoni, 2021]. Second, we notice that electrical signals stemming from segregated apical dendrites can modulate synaptic plasticity in biological neurons [Körding and König, 2001, Major et al., 2013], enabling context-dependent plasticity.

Algorithmically, our approach takes inspiration from deep self-supervised learning algorithms that seek to contrast, cluster or predict stimuli in the context of BP [Van den Oord et al., 2018, Caron et al., 2018, Zhuang et al., 2019, Löwe et al., 2019]. Interestingly, Löwe et al. [2019] demonstrated that such methods even work if end-to-end BP is partially interrupted. We build upon this body of work and suggest the *Contrastive, Local And Predictive Plasticity* (CLAPP) model which avoids BP completely, yet still builds hierarchical representations.[2]

## 2 Main goals and related work

In this paper, we propose a local plasticity rule that learns deep representations. To describe our model of synaptic plasticity, we represent a cortical area by the layer $l$ of a deep neural network. The neural activity of this layer at time $t$ is represented by the vector $\boldsymbol{z}^{t,l} = \rho(\boldsymbol{a}^{t,l})$, where $\rho$ is a non-linearity and $\boldsymbol{a}^{t,l} = \boldsymbol{W}^l \boldsymbol{z}^{t,l-1}$ is the vector of the respective summed inputs to the neurons through their basal dendrites $\boldsymbol{W}^l$ (the bias is absorbed into $\boldsymbol{W}^l$). To simplify notation, we write the pre-synaptic input as $\boldsymbol{x}^{t,l} = \boldsymbol{z}^{t,l-1}$ and we only specify the layer index $l$ when it is necessary.

Our plasticity rule exploits the fact that the temporal structure of natural inputs affects representation learning [Li and DiCarlo, 2008]. Specifically, we consider a scenario where an agent first perceives a moving object at time $t$ (e.g. a flying eagle in Figure 1 a), and then spontaneously decides to change gaze direction towards another moving object at time $t + \delta t$ (e.g. *saccade* towards the elephant in Figure 1 a). We further assume that the visual pathway is 'self-aware' of saccades due to saccade-specific modulation of processing [Ross et al., 2001].

In line with classical models of synaptic plasticity, we assume that weight changes follow biologically plausible, *Hebbian*, learning rules [Hebb, 1949, Markram et al., 2011] which are local in space and time: updates $\Delta W_{ji}^t$ of a synapse, connecting neurons $i$ and $j$, can only depend on the current activity of the pre-synaptic and post-synaptic neurons at time $t$, or slightly earlier at time $t - \delta t$, and one or several widely broadcasted modulating factors [Urbanczik and Senn, 2009, Gerstner et al., 2018].

Furthermore, we allow the activity of another neuron $k$ to influence the weight update $\Delta W_{ji}$, as long as there is an explicit connection $W_{jk}^{\text{pred}}$ from $k$ to $j$. The idea is to overcome the representational limitations of classical Hebbian learning by including dendritic inputs, which are thought to predict

---

[2]Our code is available at `https://github.com/EPFL-LCN/pub-illing2021-neurips`

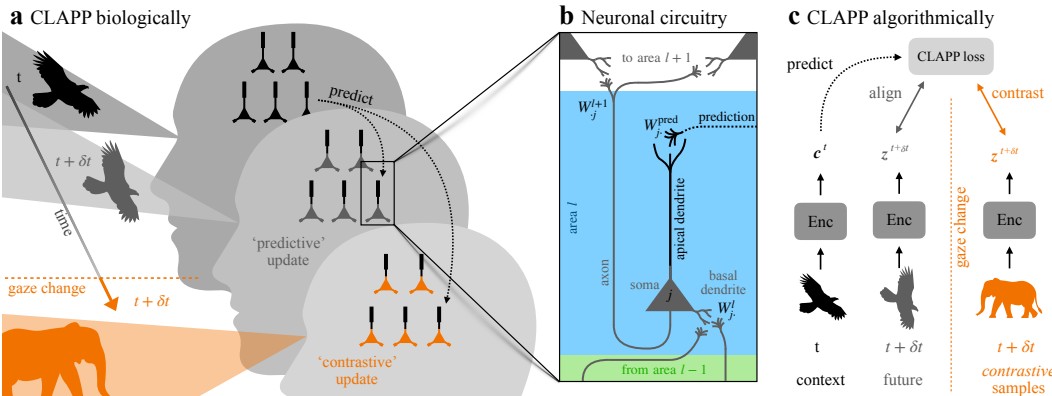

Figure 1: Contrastive, local and predictive plasticity (CLAPP). **a** Perceiving a moving object (e.g. an eagle) at times $t$ and $t + \delta t$ leads to neural responses in the visual cortex. After a gaze change ('saccade'), a different object (elephant) is seen. **b** (zoom) At each time step, pyramidal neurons integrate input activity at the basal dendrites (matrix $\boldsymbol{W}^l$ of feedforward weights) and pass on their response to downstream areas ($\boldsymbol{W}^{l+1}$). At any point in time, neurons predict future neural responses through recurrent connections $\boldsymbol{W}^{\mathrm{pred}}$. These inputs target the apical dendrites and modulate ongoing synaptic plasticity through 'predictive' updates. Information about a saccade is transmitted by a broadcast signal triggered at the moment of saccade initiation, which leads to 'contrastive' updates. As no external supervision or reward signals are provided, learning is self-supervised and local in time and space ('Hebbian'). **c** Algorithmically, an encoder network (Enc) produces a 'context' representation $\boldsymbol{c}^t$ at time $t$. Given $\boldsymbol{c}^t$, CLAPP tries to *predict* the encoding of the future input $\boldsymbol{z}^{t+\delta t}$. In case of a gaze change between $t$ and $t + \delta t$, CLAPP seeks to keep the prediction as different as possible from the encoding of the upcoming *contrastive* sample.

the future somatic activity [Körding and König, 2001, Urbanczik and Senn, 2014] and take part in the plasticity of the post-synaptic neuron [Larkum et al., 1999, Dudman et al., 2007, Major et al., 2013]. Hence we assume that each neuron $j$ in a layer $l$ may receive dendritic inputs $(\boldsymbol{W}^{\mathrm{pred}}\boldsymbol{c}^{t,l})_j$ coming either from the layer above ($\boldsymbol{c}^{t,l} = \boldsymbol{z}^{t,l+1}$) or from lateral connections in the same layer ($\boldsymbol{c}^{t,l} = \boldsymbol{z}^{t,l}$).

For algorithmic reasons, that we detail in section 3, we assume that the dendritic input $(\boldsymbol{W}^{\mathrm{pred}}\boldsymbol{c}^{t,l})_j$ influences the weight updates $\Delta W_{ji}$ of the post-synaptic neuron $j$, but not its activity $z_j^t$. This assumption is justified by neuroscientific findings that the inputs to basal and apical dendrites affect the neural activity and plasticity in different ways [Larkum et al., 1999, Dudman et al., 2007, Major et al., 2013, Urbanczik and Senn, 2014]. In general, we do not rule out influence of dendritic activity on somatic activity in later processing phases, but see this beyond the scope of the current work.

Given these insights from neuroscience, we gather the essential factors that influence synaptic plasticity in the following learning rule prototype:

$$\Delta W_{ji} \quad \propto \quad \underbrace{\text{modulators}}_{\text{broadcast factors}} \cdot \underbrace{(\boldsymbol{W}^{\mathrm{pred}}\boldsymbol{c}^{t_1})_j}_{\text{dendritic prediction}} \cdot \underbrace{\text{post}_j^{t_2} \cdot \text{pre}_i^{t_2}}_{\text{local-activity}} . \qquad (1)$$

The modulating broadcast factors are the same for large groups of neurons, for example all neurons in the same area, or even all neurons in the whole network. $\text{post}_j^{t_2}$ and $\text{pre}_i^{t_2}$ are functions of the pre- and post- synaptic activities. At this point, we do not specify the exact timing between $t_1$ and $t_2$, as this will be determined by our algorithm in section 3.

**Related work** Many recent models of synaptic plasticity fit an apparently similar learning rule prototype [Lillicrap et al., 2016, Nøkland, 2016, Roelfsema and Holtmaat, 2018, Nøkland and Eidnes, 2019, Lillicrap et al., 2020, Pozzi et al., 2020] if we interpret the top-down signals emerging from the BP algorithm as the dendritic signal. However, top-down error signals in BP are not directly related to the activity $\boldsymbol{c}^t$ of the neurons in the main network during processing of sensory input. Rather, they require a separate linear network mirroring the initial network and feeding back error signals (see Figure 2 a and Lillicrap et al. [2020]), or involved time-multiplexing of feedforward and error signals in the main network [Lillicrap et al., 2020, Payeur et al., 2021]. Our model is fundamentally different, because in our case, the dendritic signal onto neuron $j$ is strictly $(\boldsymbol{W}^{\mathrm{pred}}\boldsymbol{c}^t)_j$ which is a weighted

sum of the main network activity and there is no need of a (linear) feedback network transmitting exact error values across many layers.

Moreover, we show in simulations in section 4, that the dendritic signal does not have to come from a layer above but that the prediction fed to layer $l$ may come from the same layer. This shows that our learning rule works even in the complete absence of downward signaling from $l + 1$ to $l$. This last point is a significant difference to other methods that also calculate updates using only activities of the main network, but require tuned top-down connections to propagate signals downwards in the network hierarchy [Kunin et al., 2020], such as methods in the difference target propagation family [Lee et al., 2015, Bartunov et al., 2018, Golkar et al., 2020], contrastive divergence [Hinton, 2002] and equilibrium propagation [Scellier and Bengio, 2017]. Furthermore, the latter two require convergence to an equilibrium state for each input [Laborieux et al., 2021]. Our model does not require this convergence because it uses the recurrent dendritic signal $(\boldsymbol{W}^{\mathrm{pred}}\boldsymbol{c}^t)_j$ only for synaptic plasticity and not for inference.

Most previous learning rules which include global modulating factors interpret it as a reward prediction error [Schultz et al., 1997, Gerstner et al., 2018, Pozzi et al., 2020]. In this paper, we address self-supervised learning and view global modulating factors as broadcasting signals, modeling the self-awareness that something has changed in the stimulus (e.g. because of a saccade). Hence, the main function of the broadcast factor in our model is to identify contrastive inputs, which avoids a common pitfall for self-supervised learning models: 'trivial' or 'collapsed' solutions, where the model produces a constant output, which is easily predictable, but useless for downstream tasks. In vision, we use a broadcast factor to model the strong, saccade-specific activity patterns identified throughout the visual pathway [Kowler et al., 1995, Leopold and Logothetis, 1998, Ross et al., 2001, McFarland et al., 2015]. In other sensory pathways, like audition, this broadcast factor may model attention signals arising when changing focus on a new input source [Fritz et al., 2007], cross-modal input indicating a change in head or gaze direction, or signal/speaker-identity inferred from blind source separation, which can be done on low-level representation with biologically plausible learning rules [Hyvärinen and Oja, 1997, Ziehe and Müller, 1998, Molgedey and Schuster, 1994]. Our learning rule further requires this global factor to predict the absence or presence of a gaze change, hence conveying a *change prediction error* rather than classical reward prediction error. Here, we do not model the precise circuitry computing this factor in the brain, however, we speculate that a population of neurons could express such a scalar factor e.g. through burst-driven multiplexing of activity, see Payeur et al. [2021] and Appendix C.

Our theory takes inspiration from the substantial progress seen in unsupervised machine learning in recent years and specifically from contrastive predictive coding (CPC) [Van den Oord et al., 2018]. CPC trains a network (called *encoder*) to make *predictions* of its own responses to future inputs, while keeping this prediction as different as possible to its responses to *fake* inputs (*contrasting*). A key feature of CPC is that predicting and contrasting happens in latent space, i.e. on the output representation of the encoder network. This avoids modeling a generative model for perfect reconstruction of the input and all its details (e.g. green, spiky). Instead the model is forced to focus on extracting high-level information (e.g. cactus). In our notation, CPC evaluates a prediction $\boldsymbol{W}^{\mathrm{pred}}\boldsymbol{c}^t$ such that a score function $u_t^\tau = \boldsymbol{z}^{\tau\top}\boldsymbol{W}^{\mathrm{pred}}\boldsymbol{c}^t$ becomes larger for the true future $\tau = t + \delta t$ (referred to as positive sample) than for any other vector $\boldsymbol{z}^{t'}$ taken at arbitrary time points $t'$ elsewhere in the entire training set (referred to as negative samples in CPC). This means, that the prediction should align with the future activity $\boldsymbol{z}^{t+\delta t}$ but not with the negative samples. Van den Oord et al. [2018] formalizes this as a softmax cross-entropy classification, which leads to the traditional CPC loss:

$$\mathcal{L}_{\mathrm{CPC}}^t = -\log \frac{\exp u_t^{t+\delta t}}{\sum_{\tau \in \mathcal{T}} \exp u_t^\tau} \, , \tag{2}$$

where $\mathcal{T} = \left\{ t+\delta t, t_1' \ldots t_N' \right\}$ comprises the positive sample and $N$ negative samples. The learned model parameters are the elements of the matrix $\boldsymbol{W}^{\mathrm{pred}}$, as well as the weights of the encoder network. The loss function $\mathcal{L}_{\mathrm{CPC}}^t$ is then minimized by stochastic gradient descent on these parameters using BP. Amongst numerous recent variants of contrastive learning [He et al., 2019, Chen et al., 2020, Xiong et al., 2020], we focus here on CPC [Van den Oord et al., 2018], for which a more local variant, Greedy InfoMax, was recently proposed by Löwe et al. [2019].

Greedy InfoMax (GIM) [Löwe et al., 2019] is a variant of CPC which makes a step towards local, BP-free learning: the main idea is to split the encoder network into a few gradient-isolated modules

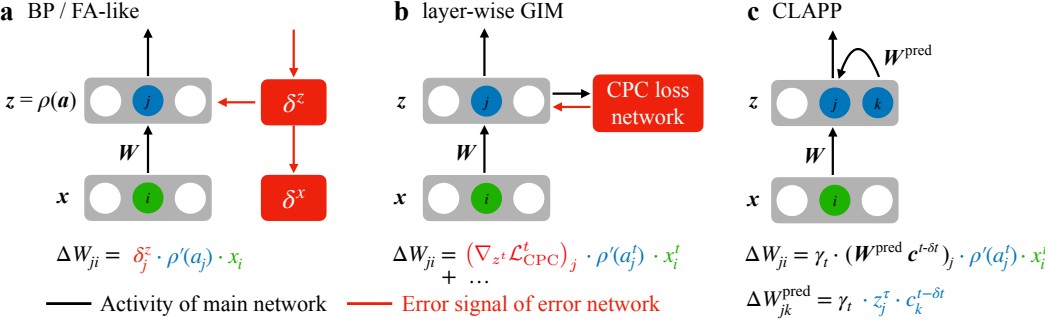

**a** BP / FA-like

$z = \rho(\boldsymbol{a})$

$\delta^z$

$\boldsymbol{W}$

$\boldsymbol{x}$

$\delta^x$

$\Delta W_{ji} = \delta_j^z \cdot \rho'(a_j) \cdot x_i$

—— Activity of main network

**b** layer-wise GIM

$z$

$\boldsymbol{W}$

$\boldsymbol{x}$

CPC loss network

$\Delta W_{ji} = \left(\nabla_{z^t}\mathcal{L}_{\mathrm{CPC}}^t\right)_j \cdot \rho'(a_j^t) \cdot x_i^t$
$\quad + \ldots$

—— Error signal of error network

**c** CLAPP

$\boldsymbol{W}^{\mathrm{pred}}$

$z$

$\boldsymbol{W}$

$\boldsymbol{x}$

$\Delta W_{ji} = \gamma_t \cdot (\boldsymbol{W}^{\mathrm{pred}}\, \boldsymbol{c}^{t-\delta t})_j \cdot \rho'(a_j^t) \cdot x_i^t$

$\Delta W_{jk}^{\mathrm{pred}} = \gamma_t \cdot z_j^\tau \cdot c_k^{t-\delta t}$

Figure 2: Comparison of weight updates **a** Networks trained with back-propagation (BP) or Feedback Alignment (FA)-like methods require separate error networks (red) for computing weight updates. **b** Layer-wise GIM, with one layer per gradient-isolated module, does not transmit error signals across layers (i.e. modules) but requires (1) the transmission of information other than the network activity (red) and (2) a perfect replay of negative samples. Thus, the resulting update computation needs a separate loss network and cannot be interpreted as a local learning rule. **c** Contrastive Local and Predictive Plasticity (CLAPP) calculates updates using locally and temporally available information: pre- and post-synaptic activity and predictive recurrent input onto the apical dendrite $\boldsymbol{W}^{\mathrm{pred}}\boldsymbol{c}^{t-\delta t}$. Global broadcasting factors $\gamma_t$ modulate plasticity depending on the presence or absence of a saccade.

to avoid back-propagation between these modules. As the authors mention in their conclusion, "*the biological plausibility of GIM is limited by the use of negative samples and within-module back-propagation*". This within-module back-propagation still requires a separate feedback network to propagate prediction errors (Figure 2 a), but can be avoided in the most extreme version of GIM, where each gradient-isolated module contains a single layer (*layer-wise GIM*). However, the gradients of layer-wise GIM, derived from Equation 2, still cannot be interpreted as synaptic plasticity rules because the gradient computation requires (1) the transmission of information other than the network activity (see Figure 2 b), and (2) perfect memory to replay the negative samples $\boldsymbol{z}^{t'}$, as mentioned in the above quote (see Appendix A for details). Overall it is not clear how this weight update of layer-wise GIM could be implemented with realistic neuronal circuits. Our CLAPP rule solves the above mentioned implausibilities and allows a truly local implementation in space and time.

## 3   Derivation of the CLAPP rule: contrastive, local and predictive plasticity

We now suggest a simpler contrastive learning algorithm which solves the issues encountered with layer-wise GIM and for which a gradient descent update is naturally compatible with the learning rule prototype from Equation 1. The most essential difference compared to CPC or GIM is, that we do not require the network to simultaneously access the true future activity $\boldsymbol{z}^{t+\delta t}$ and recall (or imagine) the network activity $\boldsymbol{z}^{t'}$ seen at some other time. Rather, we consider the naturalistic time-flow illustrated in Figure 1 a, where an agent fixates on a moving animal for a while and then changes gaze spontaneously. In this way, the prediction $\boldsymbol{W}^{\mathrm{pred}}\boldsymbol{c}^t$ is expected to be meaningful during fixation, but inappropriate right after a saccade. In our simulations, we model this by feeding the network with subsequent frames from the same sample (e.g. different views of an eagle), and then abruptly changing to frames from another sample (e.g. different views of an elephant).

We note that the future activity $\boldsymbol{z}^{t+\delta t}$ and the context $\boldsymbol{c}^t$ are *always* taken from the main feedforward encoder network. We focus on the case where the context stems from the same layer as the future activity ($\boldsymbol{c}^{t,l} = \boldsymbol{z}^{t,l}$), however, the model allows for the more general case, where the context stems from another layer (e.g. the layer above $\boldsymbol{c}^{t,l} = \boldsymbol{z}^{t,l+1}$).

**Derivation of the CLAPP rule from a self-supervised learning principle**   Rather than using a global loss function for multi-class classification to separate the true future from multiple negative samples, as in CPC, we consider here a binary classification problem at every layer $l$: we interpret the score function $u_t^{t+\delta t,l} = \boldsymbol{z}^{t+\delta t,l\top} \boldsymbol{W}^{\mathrm{pred},l} \boldsymbol{c}^{t,l}$ as the layer's 'guess' whether the agent performed a fixation or a saccade. In Appendix C, we discuss how $u_t^{t+\delta t,l}$ could be (approximately) computed in real neuronal circuits. In short, every neuron $i$ has access to its 'own' dendritic prediction $\hat{z}_i^{t,l} = \sum_j W_{ij}^{\mathrm{pred},l} c_j^{t,l}$ of somatic activity [Urbanczik and Senn, 2014], and the product $z_i^{t+\delta t,l}\, \hat{z}_i^{t,l}$

can be seen as a coincidence detector of dendritic and somatic activity, communicated by specific burst signals [Larkum et al., 1999]. These burst signals allow time-multiplexed communication [Payeur et al., 2021] of the products $z_i^{t+\delta t,l}\,\hat{z}_i^{t,l}$ of many neurons, which can then be summed by an interneuron representing $u_t^{t+\delta t,l}$.

As mentioned in section 2, information about the presence or absence of a saccade between two time points is available in the visual processing stream and is modeled here by the variable $y^t = -1$ and $y^t = +1$, respectively. We interpret $y^t$ as the label of a binary classification problem, characterized by the Hinge loss, and define the CLAPP loss at layer $l$ as:

$$\mathcal{L}_{CLAPP}^{t,l} = \max\left(0, 1 - y^t \cdot u_t^{t+\delta t,l}\right) \quad \text{with} \quad \begin{cases} y^t = +1 & \text{for fixation} \\ y^t = -1 & \text{for saccade} \end{cases} \tag{3}$$

We now derive the gradients of Equation 3 with respect to the feedforward weights and show that gradient descent on this loss function is compatible with the learning rule prototype suggested in Equation 1. Note that CLAPP optimises Equation 3 for each layer $l$ independently, without any gradient flow between layers. That being said, the following derivation is the same for every layer $l$, which is why we omit the layer index $l$ from here on.

Since we chose to formalize the binary classification with a Hinge loss, the gradient vanishes when the classification is already correct: high score $u_t^{t+\delta t} > 1$ during fixation ($y^t = +1$), or a low score $u_t^{t+\delta t} < -1$ after a saccade ($y^t = -1$). Otherwise, it is $-\nabla u_t^{t+\delta t}$ during a fixation or $\nabla u_t^{t+\delta t}$ after a saccade. In the 'predicted layer' $z$, i.e. the target of the prediction, let $W_{ji}$ denote the feedforward weight from neuron $i$ in the previous layer (with activity $x_i^t$) to neuron $j$, with summed input $a_j^t$ and activity $z_j^t$. Similarly, in the 'predicting layer' $c$, i.e. the source of the prediction, let $W_{kl}^c$ denote the feedforward weight between the neuron $l$ in the previous layer (with activity $x_l^{c,t}$) and neuron $k$, with summed input $a_k^{c,t}$ and activity $c_k^t$. Therefore, $\cdot^c$ as an upper index refers to the context layer, whereas $c$ as a full-size letter refers to the respective neuronal activity. We then find the gradients with respect to these weights as:

$$\frac{\partial \mathcal{L}_{CLAPP}^t}{\partial W_{ji}} = \pm (\boldsymbol{W}^{\mathrm{pred}} \boldsymbol{c}^t)_j\, \rho'(a_j^{t+\delta t})\, x_i^{t+\delta t} \tag{4}$$

$$\frac{\partial \mathcal{L}_{CLAPP}^t}{\partial W_{km}^c} = \pm (\boldsymbol{W}^{\mathrm{pred}\top} \boldsymbol{z}^{t+\delta t})_k\, \rho'(a_k^{c,t})\, x_m^{c,t}\,, \tag{5}$$

where the sign is negative during fixation and positive after a saccade. To change these equations into online weight updates, we consider the gradient descent update delayed by $\delta t$, such that $\Delta W_{ji}^t = -\eta \frac{\partial \mathcal{L}_{CLAPP}^{t-\delta t}}{\partial W_{ji}}$, where $\eta$ is the learning rate. Let us define a modulating factor $\gamma_t = y^t \cdot H^t$, where $y^t = \pm 1$ is a network-wide broadcast signal (self-awareness) indicating a saccade ($-1$) or a fixation ($+1$) and $H^t \in \{0, \eta\}$ is a layer-wide broadcast signal indicating whether the saccade or fixation was correctly classified as such. In this way, Equation 4 becomes a weight update which follows strictly the ideal learning rule prototype from Equation 1:

$$\Delta W_{ji}^t = \underbrace{\gamma_t}_{\text{broadcast factors}} \cdot \underbrace{(\boldsymbol{W}^{\mathrm{pred}} \boldsymbol{c}^{t-\delta t})_j}_{\text{dendritic prediction}} \cdot \underbrace{\rho'(a_j^t) x_i^t}_{\text{local activity}}\,. \tag{6}$$

For the updates of the connections onto the neuron $c_k^t$, which emits the prediction rather than receiving it, our theory in Equation 5 requires the opposite temporal order and the transmission of the information in the opposite direction: from $z^t$ back to $c^t$. Since connections in the brain are unidirectional [Lillicrap et al., 2016], we introduce another matrix $\boldsymbol{W}^{\mathrm{retro}}$ which replaces $\boldsymbol{W}^{\mathrm{pred}\top}$ in the final weight update. Given the inverse temporal order, we interpret $\boldsymbol{W}^{\mathrm{retro}} \boldsymbol{z}^t$ as a retrodiction rather than a prediction. In Appendix C, we show that using $\boldsymbol{W}^{\mathrm{retro}}$ minimises a loss function of the same form as Equation 3, and empirically performs as well as using $\boldsymbol{W}^{\mathrm{pred}\top}$. The resulting weight update satisfies the learning rule prototype from Equation 1, as it can be written:

$$\Delta W_{km}^{c,t} = \underbrace{\gamma_t}_{\text{broadcast factors}} \cdot \underbrace{(\boldsymbol{W}^{\mathrm{retro}} \boldsymbol{z}^t)_k}_{\text{dendritic retrodiction}} \cdot \underbrace{\rho'(a_k^{c,t-\delta t}) x_m^{c,t-\delta t}}_{\text{local activity}}\,. \tag{7}$$

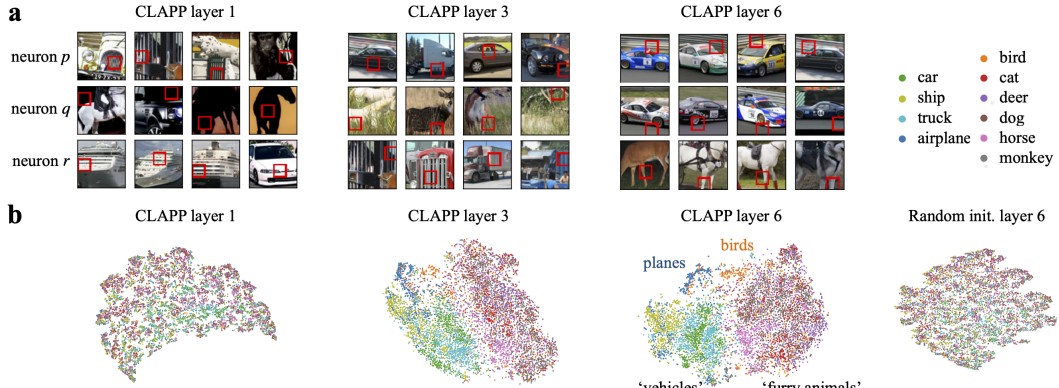

Figure 3: Hierarchical representations learned by CLAPP. **a** Red boxes in STL-10 images indicate patches that best activate a specific neuron (rows) in a network trained with CLAPP. Layer 1 extracts simple features like gratings or uniform patches, higher layers extract richer features like parts of objects. **b** 2-dimensional t-SNE projection of neuronal activities at different layers unveils increasing representational structure in higher layers (every dot represents one input image). Note that CLAPP has not seen any class labels during training.

In the (standard) case, where context and predicted activity are from the same layer ($c^{t,l} = z^{t,l}$), $W$ and $W^c$ are the same weights and the updates Equation 6 and Equation 7 are added up linearly.

The prediction and retrodiction weights, $W^{\mathrm{pred}}$ and $W^{\mathrm{retro}}$, respectively, are also plastic. By deriving the gradients of $\mathcal{L}^t_{CLAPP}$ with respect to $W^{\mathrm{pred}}$, we find an even simpler Hebbian learning rule for these weights:

$$\Delta W^{\mathrm{pred}}_{jk} = \Delta W^{\mathrm{retro}}_{kj} = \underbrace{\gamma_t}_{\text{broadcast factors}} \cdot \underbrace{z^t_j \cdot c^{t-\delta t}_k}_{\text{pre and post}}, \tag{8}$$

where neuron $k$ in the predicting layer $c$ is pre-synaptic (post-synaptic) and neuron $j$ in the predicted layer $z$ is post-synaptic (pre-synaptic) for the prediction weights $W^{\mathrm{pred}}_{jk}$ (retrodiction weights $W^{\mathrm{retro}}_{kj}$). Note that the update rules for $W^{\mathrm{pred}}_{jk}$ and $W^{\mathrm{retro}}_{kj}$ are reciprocal, a method that leads to mirrored connections, given small enough initialisation [Burbank, 2015, Amit, 2019, Pozzi et al., 2020].

We emphasize that all information needed to calculate the above CLAPP updates (Equations 6 – 8) is spatially and temporally available, either as neuronal activity at time $t$, or as traces of recent activity $(t - \delta t)$ [Gerstner et al., 2018]. In order to implement Equation 6, the dendritic prediction has to be retained during $\delta t$. However, we argue that dendritic activity can outlast (50-100 ms) somatic neuronal activity (2-10 ms) [Major et al., 2013], which makes predictive input from several time steps in the past $(t - \delta t)$ available at time $t$.

**Generalizations** While the above derivation considers fully-connected feedforward networks, we apply analogous learning rules to convolutional neural networks (CNN) and recurrent neural networks (RNN). Analyzing the biological plausibility of the standard spatial weight sharing and spatial MeanPooling operations in CNNs is beyond the scope of the current work. Furthermore, we discuss in Appendix C, that MaxPooling can be interpreted as a simple model of lateral inhibition and that gradient flow through such layers is compatibility with the learning rule prototype in Equation 1.

To obtain local learning rules even for RNNs, we combine CLAPP with the e-prop theory [Bellec et al., 2020], which provides a biologically plausible alternative to BP through time: gradients can be propagated forward in time through the intrinsic neural dynamics of a neuron using eligibility traces. The propagation of gradients across recurrently connected units is forbidden and disabled. This yields biologically plausible updates in GRU units, as explained in Appendix C.

## 4 Empirical results

**Building hierarchical representations** We first demonstrate numerically, that CLAPP yields deep hierarchical representations, despite using a local plasticity rule compatible with Equation 1. We

report here the results for $\boldsymbol{c}^{t,l} = \boldsymbol{z}^{t,l}$, i.e. the dendritic prediction in Equation 1 is generated from lateral connections and the representations in the same layer. We note, however, that we obtained qualitatively similar results with $\boldsymbol{c}^{t,l} = \boldsymbol{z}^{t,l+1}$ (i.e. the dendritic prediction is generated from one layer above), suggesting that top-down signaling is neither necessary for, nor incompatible with, our algorithm (also see Appendix C).

We first consider the STL-10 image dataset [Coates et al., 2011]. To simulate a time dimension in these static images, we follow Hénaff et al. [2019] and Löwe et al. [2019]: each image is split into $16 \times 16$ patches and the patches are viewed one after the other in a vertical order (one time step is one patch). Other hyper-parameters and data-augmentation are taken from Löwe et al. [2019], see Appendix B. We then train a 6-layer VGG-like [Simonyan and Zisserman, 2015] encoder (VGG-6) using the CLAPP rule (Equations 6 – 8). Training is performed on the unlabelled part of the STL-10 dataset for 300 epochs. We use 4 GPUs (NVIDIA Tesla V100-SXM2 32 GB) for data-parallel training, resulting in a simulation time of around 4 days per run.

In order to study how neuronal selectivity changes over layers, we select neurons randomly and show image patches which best activate these neurons the most (rows in Figure 3 a). As expected for a visual hierarchy, first-layer neurons (first column in Figure 3 a) are selective to horizontal or vertical gratings, or homogeneous colors. In the third layer of the network (second column), neurons start to be selective to more semantic features like grass, or parts of vehicles. Neurons in the last layer (third column) are selective to specific object parts (e.g. a wheel touching the road). The same analysis for a random, untrained encoder does not reveal a clear hierarchy across layers, see Appendix C.

To get a qualitative idea of the learned representation manifold, we use the non-linear dimension reduction technique t-SNE [Van der Maaten and Hinton, 2008] to visualise the encodings of the (labeled) STL-10 test set in Figure 3 b. We see that the representation in the first layer is mostly unrelated to the underlying class. In the third and sixth layers' representation, a coherent clustering emerges, yielding an almost perfect separation between furry animals and vehicles. This clustered representation is remarkable since the network has never seen class labels, and was never instructed to separate classes, during CLAPP training The representation of the same architecture, but without training (Random init.), shows that a convolutional architecture alone does not yield semantic features.

To produce a more quantitative measurement of the quality of learned representations, we follow the methodology of Van den Oord et al. [2018] and Löwe et al. [2019]: we freeze the trained encoder weights and train a linear classifier to recognize the class labels from each individual layer (Figure 4). As expected for a deep representation, the classification accuracy increases monotonically with the layer number and only saturates at layers $5$ and $6$. The accuracies obtained with layer-wise GIM are almost indistinguishable from those obtained with CLAPP. It is only at the last two layers, that layer-wise GIM performs slightly better than CLAPP; yet GIM has multiple biologically implausible features that are removed by CLAPP. As a further benchmark, we also plot the accuracies obtained with an encoder trained with greedy supervised training. This method trains each layer independently using a supervised classifier at each layer, without BP between layers, which results in an almost local update (see Löwe et al. [2019] and Appendix B). We find that accuracy is overall lower and saturates already at layer $4$. On this dataset, with many more unlabelled than labelled images, greedy supervised accuracy is almost $10\%$ below the accuracy obtained with CLAPP. Again, we see that a convolutional architecture alone does not yield hierarchical representations, as performance decreases at higher layers for a fixed random encoder.

**Comparing CPC and CLAPP**  Since CLAPP can be seen as a simplification of CPC (or GIM) we study four algorithmic differences between CPC and CLAPP individually. They are: (1) Gradients in CLAPP (layer-wise GIM) cannot flow from a layer to the next one, as opposed to BP in CPC, (2) CLAPP performs a binary comparison (fixation vs. saccade) with the Hinge loss, whereas CPC does multi-class classification with the cross entropy loss, (3) CLAPP processes a single input at a time, whereas CPC uses many positive and negative samples synchronously, and (4) we introduced $\boldsymbol{W}^{\text{retro}}$ to avoid the weight transport problem in $\boldsymbol{W}^{\text{pred}}$.

We first study features (1) and (2) but relax constraints (3) and (4). That means, in this paragraph, we allow a fixation and $N = 16$ synchronous saccades and set $\boldsymbol{W}^{\text{retro}} = \boldsymbol{W}^{\text{pred},\top}$. We refer to *Hinge Loss CPC* as the algorithm minimizing the CLAPP loss (Equation 3) but using end-to-end BP. *CLAPP-s* (for *synchronous*) applies the Hinge Loss to every layer, but with gradients blocked between layer. We find that the difference between the CPC loss and Hinge Loss CPC is less than $1\%$, see

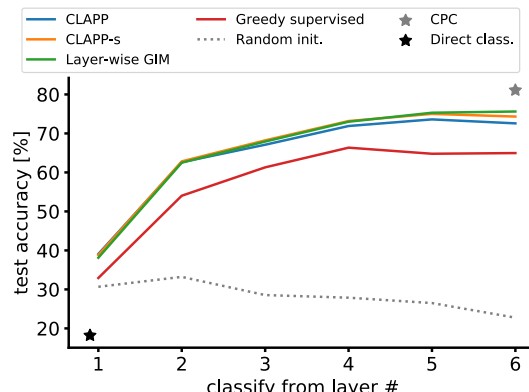

Figure 4: CLAPP stacks well: representations after stacking up to 5 layers increase performance of a linear classifier on STL-10, despite the local learning rule (blue and orange lines), while performance decreases for convolutional network with weights fixed at random initialisation (dotted). Greedy supervised training (see Appendix B) also stacks, but already saturates at layer 4 and shows overall lower performance. Direct linear classification on image pixels (black star) and CPC performance after 6 layers (gray star) serve as upper and lower performance bounds, respectively.

Table 1. In contrast, additional blocking of gradients between layers causes a performance drop of almost 5% for both loss functions. We investigate how gradient blocking influences performance with a series of simulations, splitting the 6 layers of the network into two or three gradient isolated modules, exploring the transition from Hinge Loss CPC to CLAPP-s. Performance drops monotonously but not catastrophically, as the number of gradient blocks increases (Table 1).

CLAPP's temporal locality allows the interpretation that an agent alternates between fixations and saccades, rather than perfect recall and synchronous processing of negative samples, as required by CPC. To study the effect of temporal locality, we apply features (2) and (3) and relax the constraints (1) and (4). The algorithm combining temporal locality and the CLAPP loss function is referred to as *time-local Hinge Loss CPC*. We find that the temporal locality constraint decreases accuracy by 1.2% compared to Hinge Loss CPC. The last feature introduced for biological plausibility is using the matrix $W^{\text{retro}}$ and we observe almost no difference in classification accuracy with this alternative (the accuracy decreases by 0.1%). Conversely, omitting the update in Equation 7 entirely, i.e. setting the retrodiction $W^{\text{retro}} = 0$, compromises accuracy by 2% compared to vanilla Hinge Loss CPC.

When combining all features (1) to (4), we find that the fully local CLAPP learning rule leads to an accuracy of 73.6% at layer 5. We conclude from the analysis above, that the feature with the biggest impact on performance is (1): blocking the gradients between each layer. However, despite the performance drop caused by blocking the gradients, CLAPP still stacks well and leverages the depth of the network (Figure 4). All other features (2) - (4), introduced to derive a weight update compatible with our prototype (Equation 1), only caused a minor performance loss.

**Applying CLAPP to speech and video**    We now demonstrate that CLAPP is applicable to other modalities like the LibriSpeech dataset of spoken speech [Panayotov et al., 2015] and the UCF-101 dataset containing short videos of human actions [Soomro et al., 2012]. When applying CLAPP to auditory signals, we do not explicitly model the contrasting mechanism (saccades in the vision task; see discussion in section 2 for the auditory pathway) and hence consider the application of CLAPP as benchmark application, rather than a neuroscientifically exact study. To increase computational efficiency, we study CLAPP-s on speech and video. Based on the image experiments, we expect similar results for CLAPP, given enough time to converge. We use the same performance criteria as for the image experiments and summarize our results in Table 1, for details see Appendix B.

For the audio example, we use the same architecture as Van den Oord et al. [2018] and Löwe et al. [2019]: multiple temporal 1d-convolution layers and one recurrent GRU layer on top. As in the feedforward case, CLAPP still optimises the objective of Equation 3. For the 1d-convolution layers, the context $c^t$ is computed as for the image task, for the last layer, $c^t$ is the output of the recurrent GRU layer. We compare the performance of the algorithms on phoneme classification (41 classes) using labels provided by Van den Oord et al. [2018]. In this setting, layer-wise training lowers performance by only 0.4% for layer-wise GIM, and by 1.1% for CLAPP-s. Implemented as such, CLAPP-s still relies on BP through time (BPTT) to train the GRU layer. Using CLAPP-s with biologically plausible e-prop [Bellec et al., 2020], instead of non-local BPTT, reduces performance by only 3.1 %, whereas omitting the GRU layer compromises performance by 9.3 %, see Appendix C.

Applying CLAPP to videos is especially interesting because their temporal sequence of images perfectly fits the scenario of Figure 1 a. In this setting, we take inspiration from Han et al. [2019],

Table 1: CLAPP performs best among methods that are local in space and time. Linear classification test accuracy [%] on STL-10, phone classification on LibriSpeech, and video human action recognition on UCF-101 using features from the encoder trained with different methods. On STL-10, performance degrades gracefully with the number of gradient-isolated modules in the VGG-6 encoder (at fixed number of encoder layers). Greedy supervised training uses BP in auxiliary classifier networks ('almost' local in space). For LibriSpeech, BP through time is used (can be avoided, see Appendix C). Values with * are taken from Löwe et al. [2019]. For simulation details, see Appendix B.

| Method | local in ... space? | time? | STL-10 | LibriSpeech | UCF-101 |
|---|---|---|---|---|---|
| Chance performance | | | 10.0 | 2.4 | 0.99 |
| Random init. | ✓ | ✓ | 21.8 | 27.7* | 30.5 |
| MFCC | ✓ | ✓ | - | 39.7* | - |
| Greedy supervised | (✓) | ✓ | 66.3 | 73.4* | - |
| Supervised | ✗ | ✓ | 73.2 | 77.7* | 51.5 |
| CPC | ✗ | ✗ | 81.1 | 64.3 | 35.7 |
| Layer-wise GIM | ✗ | ✗ | 75.6 | 63.9 | 41.2 |
| Hinge Loss CPC (ours) | ✗ | ✗ | 80.3 | 62.8 | 36.1 |
| CLAPP-s (2 modules of 3 layers) | ✗ | ✗ | 77.6 | - | - |
| CLAPP-s (3 modules of 2 layers) | ✗ | ✗ | 77.4 | - | - |
| CLAPP-s (ours) | ✓ | ✗ | 75.0 | 61.7 | 41.6 |
| time-local Hinge Loss CPC (ours) | ✗ | ✓ | 79.1 | - | - |
| CLAPP (ours) | ✓ | ✓ | 73.6 | - | - |

and use a VGG-like stack of 2D and 3D convolutions to process video frames over time. On this task (101 classes), we found layer-wise GIM and CLAPP-s to achieve higher downstream classification accuracy than their end-to-end counterparts CPC and Hinge Loss CPC (see Table 1), in line with the findings on STL-10 in Löwe et al. [2019]. On the other hand, we found that CLAPP-s requires more negative samples (i.e. more simultaneous comparisons of positive and negative samples) on videos than on STL-10 and LibriSpeech. Under the constraint of temporal locality in fully local CLAPP, this leads to prohibitively long convergence times in the current setup. However, since CLAPP linearly combines updates stemming from multiple negative and positive samples, we eventually expect the same final performance, if we run the online CLAPP algorithm for a sufficiently long time.

## 5    Discussion

We introduced CLAPP, a self-supervised and biologically plausible learning rule that yields deep hierarchical representations in neural networks. CLAPP integrates neuroscientific evidence on the dendritic morphology of neurons and takes the temporal structure of natural data into account. Algorithmically, CLAPP minimises a layer-wise contrastive predictive loss function and stacks well on different task domains like images, speech and video – despite the locality in space and time.

While the performance loss due to layer-wise training is a limitation of the current model, the stacking property is preserved and preliminary results suggest improved versions that stack even better (e.g. using adaptive encoding patch sizes). Note that CLAPP models self-supervised learning of cortical hierarchies and does *not* provide a general credit assignment method, such as BP. However, the representation learned with CLAPP could serve as an initialisation for transfer learning, where the encoder is fine-tuned later with standard BP. Alternatively, fine-tuning could even start already during CLAPP training. CLAPP in its current form is data- and compute-intensive, however, it runs on unlabelled data with quasi infinite supply, and is eligible for neuromorphic hardware, which could decrease energy consumption dramatically [Wunderlich et al., 2019].

Classical predictive coding models alter neural activity at inference time, e.g. by cancelling predicted future activity [Rao and Ballard, 1999, Keller and Mrsic-Flogel, 2018]. Here, we suggest a different, perhaps complementary, role of predictive coding in synaptic plasticity, where dendritic activity predicts future neural activity, but directly enters the learning rule [Körding and König, 2001, Urbanczik and Senn, 2014]. CLAPP currently does not model certain features of biological neurons, e.g. spiking activity or long range feedback, and requires neurons to transmit signals with precise value and timing. We plan to address these topics in future work.

## Acknowledgments and Disclosure of Funding

This research was supported by the Swiss National Science Foundation (no. 200020_184615) and the Intel Neuromorphic Research Lab. Many thanks to Sindy Löwe, Julie Grollier, Maxence Ernoult, Franz Scherr, Johanni Brea and Martin Barry for helpful discussions. Special thanks to Sindy Löwe for publishing the GIM code.

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
