**Appendices of:**

---

# Local plasticity rules can learn deep representations using self-supervised contrastive predictions

---

**Notation in appendices**  In all appendices, and in line with [Van den Oord et al., 2018, Löwe et al., 2019], the context vector, from which the prediction is performed, is denoted $\boldsymbol{c}^t$ and the feature vector being predicted is denoted $\boldsymbol{z}^{t+\delta t}$ (or $\boldsymbol{z}^{t'}$ for negative samples). In general, the loss function of CPC and CLAPP are therefore defined with the score functions $u_t^\tau = \boldsymbol{z}^{\tau\,\top} \boldsymbol{W}^{\mathrm{pred}} \boldsymbol{c}^t$.

Throughout the vision experiments and when training the temporal convolutions of the audio processing network, it happens that $\boldsymbol{c}$ and $\boldsymbol{z}$ denote the same layer (see Appendix B for details). However, when processing audio, the highest loss uses the last layer as the context layer $\boldsymbol{c}$ and the one before last for $\boldsymbol{z}$.

To cover the most general case, we introduce different notations for the parameters and the variables of the context layer $\boldsymbol{c}$ and the feature layer $\boldsymbol{z}$. For simplicity our analysis considers standard, fully-connected networks – even if the reasoning generalises easily to other architectures. Hence, with a non-linearity $\rho$, the feature layer produces the activity $\boldsymbol{z}^t = \rho(\boldsymbol{a}^{\boldsymbol{z},t})$ with $\boldsymbol{a}^{\boldsymbol{z},t} = \boldsymbol{W}^{\boldsymbol{z}} \boldsymbol{x}^{\boldsymbol{z},t} + \boldsymbol{b}^{\boldsymbol{z}}$ where $\boldsymbol{x}^{\boldsymbol{z},t}$, $\boldsymbol{W}^{\boldsymbol{z}}$ and $\boldsymbol{b}^{\boldsymbol{z}}$ are the input vector (at time $t$), weight matrix and bias respectively (the layer index $l$ is omitted for simplicity). The notation naturally extends to the context layer $\boldsymbol{c}$ and we use $\boldsymbol{x}^{\boldsymbol{c},t}$, $\boldsymbol{W}^{\boldsymbol{c}}$ and $\boldsymbol{b}^{\boldsymbol{c}}$ to denote its input and its parameters. Note that when the context and feature layer are the same layer $\boldsymbol{z} = \boldsymbol{c}$, the two parameters $\boldsymbol{W}^{\boldsymbol{c}}$ and $\boldsymbol{W}^{\boldsymbol{z}}$ are actually only one single parameter $\boldsymbol{W}$ and the weight update is given by $\Delta \boldsymbol{W} = \Delta \boldsymbol{W}^c + \Delta \boldsymbol{W}^z$.

For the gradient computations in the appendices we assume that the gradient cannot propagate further than one layer. Hence, $\boldsymbol{x}^z$ and $\boldsymbol{x}^c$ are always considered as constants with respect to all parameters, even though this is technically not true, for instance with $\boldsymbol{c}^l = \boldsymbol{z}^{l+1}$. In this case we would have $\boldsymbol{z} = \boldsymbol{x}^{\boldsymbol{c}}$ and thus $\nabla_{\boldsymbol{W}^{\boldsymbol{z}}} \boldsymbol{x}^{\boldsymbol{c}} \neq \boldsymbol{0}$, but we use the convention $\nabla_{\boldsymbol{W}^{\boldsymbol{z}}} \boldsymbol{x}^{\boldsymbol{c}} = \boldsymbol{0}$ to obtain local learning rules. Gradients are computed accordingly by stopping gradient propagation in all our experiments.

## A  Analysis of the original CPC gradient

Even after preventing gradients to flow from a layer to the next, we argue that parts of the gradient computation in CPC and GIM are hard to implement with the type of information processing that is possible in neural circuits. For this reason we analyse the actual gradients computed by layer-wise GIM. We further discuss the bio-plausibility of the resulting gradient computation in this section.

To derive the loss gradient we define the probability $\pi_t^{t*}$ that the sample $\boldsymbol{z}^{t*}$ is predicted as the true future given the context layer $\boldsymbol{c}^t$: $\pi_t^{t*} \overset{\mathrm{def}}{=} \frac{1}{\mathcal{Z}} \exp u_t^{t*}$ with $\mathcal{Z} \overset{\mathrm{def}}{=} \sum_{\tau \in \mathcal{T}} \exp u_t^\tau$. The set $\mathcal{T} = \left\{ t^{t+\delta t}, t_1' \ldots t_N' \right\}$ comprises the positive and $N$ negative samples. We have in particular $\mathcal{L}_{CPC}^t = -\log \pi_t^{t+\delta t}$ and for any parameter $\theta$ the (negative) loss gradient is given by:

$$\nabla_\theta \log \pi_t^{t+\delta t} = \nabla_\theta u_t^{t+\delta t} - \sum_{\tau \in \mathcal{T}} \pi_t^\tau \, \nabla_\theta u_t^\tau \; . \tag{9}$$

We consider only three types of parameters: the weights $\boldsymbol{W}^c$ onto the context vector $\boldsymbol{c}^t$, the weights $\boldsymbol{W}^z$ onto the feature vector $\boldsymbol{z}^{t*}$ and the weights $\boldsymbol{W}^{\mathrm{pred}}$ defining the scalar score $u_t^{t*} = \boldsymbol{z}^{t*\,\top} \boldsymbol{W}^{\mathrm{pred}} \boldsymbol{c}^t$ (the biases are absorbed in the weight matrices for simplicity).

Let's first analyze the gradient with respect to $\boldsymbol{W}^{\mathrm{pred}}$. Using the conventions that $k$ is the index of the context unit $c_k$ and $j$ is the index of the feature unit $z_j$, we have:

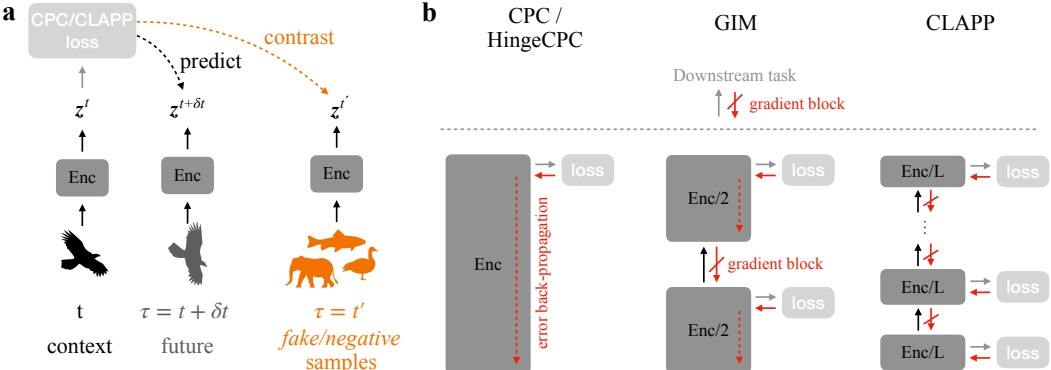

Figure 5: **a** In Contrastive Predictive Coding (CPC) and CLAPP(-s), an encoder network (Enc) produces a representation $z^t$ at time $t$ (sometimes more generally called 'context'). Given $z^t$, the encoding of the future input $z^{t+\delta t}$ should be *predicted* while keeping the prediction as different as possible from encoded *fake* or *negative* samples $z^{t'}$ (*contrasting*). The loss function implementing this contrasting depends on the method: CPC uses cross-entropy classification, CLAPP uses a Hinge-loss. **b** CPC trains the encoder network end-to-end using gradient back-propagation (red arrows). In Greedy InfoMax (GIM), the encoder network is split into several, gradient-isolated modules and the loss (CPC or Hinge) is applied separately to each module. Gradient back-propagation still occurs within modules (red, dashed arrows) but is blocked between modules. In CLAPP, every module contains only a single trainable layer of the $L$-layer encoder. This avoids any back-propagation and makes CLAPP layer-local.

$$\nabla_{W_{jk}^{\mathrm{pred}}} \log \pi_t^{t+\delta t} = c_k^t \left( z_j^{t+\delta t} - \sum_{\tau \in \mathcal{T}} \pi_t^{\tau} \, z_j^{\tau} \right) \tag{10}$$

Viewing a gradient descent weight update of that parameter as a model of synaptic plasticity in the brain raises essential questions. If $z_j^{t+\delta t} - \sum_{\tau \in \mathcal{T}} \pi_t^{\tau} z_j^{\tau}$ was the activity of the unit $j$, it would boil down to a Hebbian learning rule, well supported experimentally, but the activity of unit $j$ is considered to be the vector element $z_j$ since it is transmitted to the layer above during inference. Hence, the unit $j$ would have to transmit two distinct quantities at the same time, which is unrealistic when modelling real neurons. On top of that, it is unclear how the term $\sum_{\tau \in \mathcal{T}} \pi_t^{\tau} z_i^{\tau}$ would be computed.

We now compute the gradient with respect to $W^c$ and $W^z$. The update of these parameters raises an extra complication because it involves the activity of more than two units. For the parameters of the layer $z$ we denote $j$ a neuron in this layer, and $i$ a neuron from its input layer $x$. Then the loss gradient is given by:

$$\nabla_{W_{ji}^z} \log \pi_t^{t+\delta t} = (W^{\mathrm{pred}} c^t)_j \left( \rho'(a_j^z)^{t+\delta t} x_i^{z,t+\delta t} - \sum_{\tau \in \mathcal{T}} \pi_t^{\tau} \, \rho'(a_j^z)^{\tau} x_i^{z,\tau} \right) . \tag{11}$$

Similarly, for the parameters of a neuron $c_j^t$:

$$\nabla_{W_{ji}^c} \log \pi^{t+\delta t} = \left( W^{\mathrm{pred},\top} \left( z^{t+\delta t} - \sum_{\tau \in \mathcal{T}} \pi_t^{\tau} \, z^{\tau} \right) \right)_j \rho'(a_j^c)^t x_i^{c,t} . \tag{12}$$

These gradients raise the same essential problems as the computation of the gradients with respect to $W^{\mathrm{pred}}$ and even involve other complex computations.

## B   Simulation details

We use pytorch [Paszke et al., 2017] for our implementation and base it on the code base of the GIM paper [Löwe et al., 2019] [3]. Unless mentioned otherwise we adopt their setup, data sets, data handling and (hyper-)parameters.

---

[3] https://github.com/loeweX/Greedy_InfoMax

### B.1 Vision experiments

**General procedure**  We use the STL-10 dataset, designed for unsupervised learning algorithms [Coates et al., 2011], which contains $100,000$ unlabeled color images of $96 \times 96$ pixels. Additionally, STL-10 contains a much smaller labeled training set with 10 classes and 500 training images per class and 800 labeled test images per class. Since CPC-like methods rely on sequences of data we have to introduce an artificial 'temporal' dimension in the case of vision data sets. To simulate a time dimension in these static images we represent the motion of the visual scene by splitting the image into partially overlapping tiles. Then, vertical slices of patches define a temporal order, as in Hénaff et al. [2019] and Löwe et al. [2019]: the patches are viewed one after the other in a vertical order (one time step is one patch). The hyper-parameters of this procedure and of any other image preprocessing and data augmentation steps are as in Löwe et al. [2019].

This results in a time varying input stimulus which is fed into the encoder network and the weights of this network are updated using the CLAPP rule Equation 6 and Equation 7 (or reference algorithms, respectively). CLAPP represents saccades towards a new object by changing the next input image to a different one at any time step with probability $0.5$. Note that this practice reduces the number of training data by $50\%$ compared to CLAPP-s, GIM and CPC, which are updated with positive and negative sample synchronously at every step. Since this slows down convergence, we grant CLAPP double the amount of training epochs to yield a fair comparison ($1\%$ improvement for CLAPP). We leverage common practices from deep learning to accelerate the simulation: the weight changes are averaged and applied after going through a batch of 32 images so that the images can be processed in parallel. We accumulate the gradient updates and use the Adam optimiser with fixed learning rate $0.0002$.

We then freeze the encoder network and train a linear downstream classifier on representations created by the encoder using held-out, labeled data from 10 different classes from the STL-10 dataset. The accuracy of that classification serves as a measure to evaluate the quality of the learned encoder representations.

**Encoder architecture**  We use VGG-6, a custom 6-layer VGG-like [Simonyan and Zisserman, 2015] encoder with 6 trainable layers (6 convolutional, 4 MaxPool, 0 fully-connected, see Table 2). The architecture choice was inspired by the condensed VGG-like architectures successfully applied in Nøkland and Eidnes [2019]. The main motivation was to work with an architecture that allows pure layer-wise training which is impossible in e.g. ResNet-50 due to skip-connections. Surprisingly we find that the transition from ResNet-50 to VGG-6 does neither compromise CPC losses nor downstream classification performance for almost all training methods, see Table 4.

| # of trainable layer | layer type |
|:---:|:---:|
| 1 | 3×3 conv128, ReLU |
| 2 | 3×3 conv256, ReLU |
|   | 2×2 MaxPool |
| 3 | 3×3 conv256, ReLU |
| 4 | 3×3 conv512, ReLU |
|   | 2×2 MaxPool |
| 5 | 3×3 conv1024, ReLU |
|   | 2×2 MaxPool |
| 6 | 3×3 conv1024, ReLU |
|   | 2×2 MaxPool |

Table 2: Architecture of the VGG-6 encoder network. Convolutional layers (conv) have stride (1, 1), Pooling layers use stride (2, 2). The architecture is inspired by the VGG-like networks used in Nøkland and Eidnes [2019].

In GIM and CLAPP, the encoder is split into several, gradient-isolated modules. Depending on the number of such modules, each module contains a different number of layers. In CPC we do not use any gradient blocking and consequently the encoder consists only of one module containing layers 1-6. In layer-wise GIM and CLAPP each of the 6 modules contains exactly on layer (and potentially

another MaxPooling layer). Table 3 shows the distribution of layers into modules for the cases in between.

Table 3: Distribution of layers into modules as done for the simulations in Table 1 and Table 6. The layer numbers refer to Table 2.

| # of modules | layer distribution |
|---|---|
| 1 (CPC) | (1,2,3,4,5,6) |
| 2 | (1,2,3), (4,5,6) |
| 3 | (1,2), (3,4), (5,6) |
| 4 | (1,2,3),(4),(5),(6) or (1),(2),(3),(4,5,6) |
| 6 | (1),(2),(3),(4),(5),(6) |

Table 4: Linear classification test accuracy (%) on STL-10 with features coming from two different encoder models: ResNet-50 as in Löwe et al. [2019] and a 6-layer VGG-like encoder (VGG-6). Values for ResNet-50 are taken from Löwe et al. [2019].

|  | ResNet-50 | VGG-6 |
|---|---|---|
| Random init | 27.0 | 21.8 |
| Greedy Supervised | 65.2 | 65.0 |
| Supervised | 71.4 | 73.2 |
| CPC | 80.5 | 81.1 |
| GIM (3 modules) | **81.9** | 78.3 |

**Reference algorithms**  *Random init* refers to the random initialisation of the encoder network. It thus represents an untrained network with random weight matrices. This 'method' serves as a lower bound on performance and as a sanity check for other algorithms.

In classic *supervised* training, we add a fully-connected layer with as many output dimensions as classes in the data set to the encoder architecture. Then the whole stack is trained end-to-end using a standard supervised loss and back-propagation. For data sets offering many labels this serves as an upper bound on performance of unsupervised methods. In the case of sparsely labeled data, unsupervised learners could, or even should, outperform supervised learning.

The *greedy supervised* method trains every gradient-isolated module of the encoder separately. For that, one fully-connected layer is added on top of each module. Then, for every module, the stack consisting of the module and the added fully-connected layer is trained with a standard supervised loss requiring labels. Gradients are back-propagated within the module but blocked between modules. This layer-wise training makes the method quasi layer-local, however, BP through the added fully-connected layer is still required.

### B.2  Audio experiments

We follow most of the implementation methods used in Löwe et al. [2019]. The model is trained without supervision on 100 hours of clean spoken sentences from the LibriSpeech data set [Panayotov et al., 2015] without any data augmentation. For feature evaluation, a linear classifier is used to extract the phonemes divided into 41 classes. This classifier is trained on the test split of the same dataset, along with the phoneme annotations computed with a software from Van den Oord et al. [2018].

The audio stream is first processed with four 1D convolutional layers and one recurrent layer of Gated Recurrent Units (GRU). The hyperparameters of this architecture are the same as the ones used in Löwe et al. [2019].

All convolutional layers are assigned a CPC or a CLAPP loss as described in the main text and the gradients are blocked between them. To train the last layer – the recurrent layer –, we add one variant of the CLAPP and CPC losses where the score function is defined by $u_t^\tau = z^{\tau\top} W^{\mathrm{pred}} c^t$ where $c^t$ is the activity of the GRU layer and $z^\tau$ is the activity of the last layer of convolutions. This loss is minimized with respect to the parameters of $c$ and $z$, and the gradients cannot flow to the layers below (hence $\nabla_{W^z} c^t = 0$ even if $z$ is implicitly the input to $c$ with this architecture).

Within the GRU layer the usual implementation of gradient descent with pytorch involves back-propagation through time (BPTT), even if we avoided BP between layers. To avoid all usage of back-propagation and obtain a more plausible learning rule we used e-prop [Bellec et al., 2020] instead of BPTT. The details of this implementation are provided in the next section (Appendix C) in the paragraph 'Combining e-prop and CLAPP'.

### B.3 Video experiments

**General procedure**   We use the UCF-101 dataset [Soomro et al., 2012], an action recognition dataset containing 13,000 videos representing 101 actions. The original clips have a frequency of 30 frames per second and were downsampled by a factor 3. Videos were cut into clips of respectively 54 frames (5.4 seconds) for self-supervised learning and 72 (7.2 seconds) for the following classification. Frames in a clip were randomly grayed and jittered following the procedure of Han et al. [2019]. Cropping and horizontal flipping were applied per clip.

**Architecture and training**   For our network, we use a VGG-like network with 5 trainable layers presented in table 5. The architecture is decomposed into spatial convolutions processing frames individually and additional temporal convolutions accounting for the temporal component of a clip. The first convolution uses no padding and all others have padding (0, 1, 1). The stride used for the spatial convolutions is, respectively, (1, 2, 2), (1, 2, 2) and (1, 1, 1) whereas the temporal convolutions both have stride (3, 1, 1) to prevent temporal overlap between successive encodings.

Table 5: Architecture of the VGG-5 encoder network. Pooling layers use stride (1, 2, 2).

| # of trainable layer | layer type |
|:---:|:---:|
| 1 | $1\times7\times7$ conv96, BN, ReLU |
|   | $1\times3\times3$ MaxPool |
| 2 | $1\times5\times5$ conv256, BN, ReLU |
|   | $1\times3\times3$ MaxPool |
| 3 | $1\times3\times3$ conv512, BN, ReLU |
| 4 | $3\times3\times3$ conv512, BN, ReLU |
| 5 | $3\times3\times3$ conv512, BN, ReLU |
|   | $1\times3\times3$ MaxPool |

Whereas Löwe et al. [2019] applies pooling to the feature maps outputted by a layer to obtain the encoding, we flatten them to preserve spatial information necessary to understand and predict the spatial flow and structure from movements related to an action.

For the training procedure, we use a batch size of 8 and train for 300 epochs with a fixed learning rate of 0.001. We use as many negative samples as available in the batch, for the spatial convolutions this leads to 429 negatives and the two temporal convolutions respectively have 141 and 45. This decrease is due to the temporal reductions occurring, aimed at preventing information leakage between sequences.

## C   Additional material

**Weight transport in $W^{\mathrm{pred}}$**   The update of the encoder weights $\mathbf{W}$ in CPC, GIM and CLAPP (before introducing $W^{\mathrm{retro}}$) relies on weight transport in $W^{\mathrm{pred}}$, as seen in Equation 12 or Equation 5.

The activity of $c^t$ is propagated with the matrix $W^{\mathrm{pred}}$ and $z^\tau$ with its transpose. This is problematic because typical synapses in the brain transmit information only in a single direction. The existence of a symmetric reverse connection matrix would solve this problem but raises the issue that connection strengths would have to be synchronised (hence the word *weight transport*) between $W^{\mathrm{pred}}$ and the reverse connections.

One first naive solution is to block the gradient at the layer $c$ in the definition of the score $u_t^\tau = z^{\tau\top} W^{\mathrm{pred}} \mathrm{block\_grad}(c^t)$, with the definition:

$$
\begin{aligned}
\mathrm{block\_grad}(x) &= x \\
\nabla_x \mathrm{block\_grad}(x) &= 0 \,.
\end{aligned}
\tag{13}
$$

In this way, no information needs to be transmitted through the transpose of $W^{\mathrm{pred}}$. However this results in a relatively large drop in performance on STL-10 for Hinge Loss CPC (78.0 %) and CLAPP (70 %).

A better option – and as done in the main paper – is to split the original $W^{\mathrm{pred}}_{\mathrm{orig}}$ into two matrices $W^{\mathrm{pred}}$ and $W^{\mathrm{retro}}$ (for 'retrodiction') which are independent and which allow information flow only in a single direction (as in actual biological synapses). To this end, we split the loss function into two parts: one part receives the activity $W^{\mathrm{pred}} c^t$ coming from $c^t$ and only updates the parameters of $z$; and the other part receives the activity $W^{\mathrm{retro}} z^\tau$ coming from $z^\tau$ and updates the parameters of $c$. Like this information is transmitted through $W^{\mathrm{pred}}$ and $W^{\mathrm{retro}}$ instead of $W^{\mathrm{pred}}$ and its transpose matrix and hence solves the weight transport problem.

More formally, let us write $F$ to summarize the definition of the usual CLAPP loss function in Equation 3 such that $\mathcal{L}^t_{\mathrm{CLAPP}} = F(c^t, z^\tau, W^{\mathrm{pred}}_{\mathrm{orig}})$. We then introduce a modified version of the CLAPP loss function:

$$
\tilde{\mathcal{L}}^t_{\mathrm{CLAPP}} = \frac{1}{2}\left( \tilde{\mathcal{L}}^{t,z}_{\mathrm{CLAPP}} + \tilde{\mathcal{L}}^{t,c}_{\mathrm{CLAPP}} \right) \,,
\tag{14}
$$

with $\tilde{\mathcal{L}}^{t,z}_{\mathrm{CLAPP}} = F(\mathrm{block\_grad}(c^t), z^\tau, W^{\mathrm{pred}})$ and $\mathcal{L}^{t,c}_{\mathrm{CLAPP}} = F(c^t, \mathrm{block\_grad}(z^\tau), W^{\mathrm{retro}})$. Similarly, we define the corresponding scores as $u_t^{\tau,z} = z^{\tau\top} W^{\mathrm{pred}} \mathrm{block\_grad}(c^t)$ and $u_t^{\tau,c} = \mathrm{block\_grad}(z^\tau)^\top W^{\mathrm{retro},\top} c^t$. With this, the gradients with respect to the weight parameters $W^z_{ji}$ (encoding $z^\tau$) are:

$$
\frac{\partial u_t^{\tau,z}}{\partial W^z_{ji}} = x_i^{\tau,z} \rho'(a_j^{\tau,z})(W^{\mathrm{pred}} c^t)_j \qquad \text{and} \qquad \frac{\partial u_t^{\tau,c}}{\partial W^z_{ji}} = 0 \,,
\tag{15}
$$

and the gradients with respect to the weights $W^c_{ji}$ (encoding $c^t$) become:

$$
\frac{\partial u_t^{\tau,z}}{\partial W^c_{ji}} = 0 \qquad \text{and} \qquad \frac{\partial u_t^{\tau,c}}{\partial W^c_{ji}} = x_i^{t,c} \rho'(a_j^{t,c})(W^{\mathrm{retro}} z^\tau)_j \,.
\tag{16}
$$

The final plasticity rule combines those terms and recovers the original CLAPP rule Equation 6 and Equation 7:

$$
\begin{aligned}
\Delta W^\tau_{ji} &= \gamma_\tau \left[ \Delta W^{\tau,z}_{ji} + \Delta W^{\tau,c}_{ji} \right] \\
\Delta W^{\tau,z}_{ji} &= \left( W^{\mathrm{pred}} c^t \right)_j \rho'(a_j^{\tau,z}) x_i^{\tau,z} \\
\Delta W^{\tau,c}_{ji} &= \left( W^{\mathrm{retro}} z^\tau \right)_j \rho'(a_j^{t,c}) x_i^{t,c},
\end{aligned}
\tag{17}
$$

under the assumption of having only one gating factor $\gamma_\tau$. This is approximately the case when $W^{\mathrm{pred}}$ and $W^{\mathrm{retro}}$ align since then $u_t^\tau = u_t^{\tau,c} = u_t^{\tau,z}$. We consider this assumption realistic since $W^{\mathrm{pred}}$ and $W^{\mathrm{retro}}$ share the same update rule Equation 8. We see that the propagation of the activity through the independent weights $W^{\mathrm{pred}}$ and $W^{\mathrm{retro}}$ is always unidirectional.

It turns out that, using the modified loss $\tilde{\mathcal{L}}^t_{\mathrm{CLAPP}}$, Equation 14, instead of the original CLAPP loss $\mathcal{L}^t_{\mathrm{CLAPP}}$, Equation 3, the performance on STL-10 (linear classification on last layer) is unchanged for Hinge Loss CPC (80.2 %) and CLAPP-s (74.1 %).

**Combining e-prop and CLAPP** CLAPP avoids the usage of back-propagation through the depth of the network, but when using a recurrent GRU layer in the audio task, gradients are still back-propagated through time inside the layer. A more plausible alternative algorithm has been suggested in Bellec et al. [2020]: synaptic eligibility traces compute local gradients forward in time using the activity of pre- and post-synaptic units, then these traces are merged with the learning signal (here $W z^{t+\delta t}$) to form the weight update. It is simple to implement e-prop with an auto-differentiation

software such as pytorch by introducing a block_grad function in the update of the recurrent network. With GRU, we implement a custom recurrent network as follows (the notations are consistent with the pytorch tutorial on GRU networks[4] and unrelated to the rest of the paper):

$$r_t = \sigma(W_{ir}x_t + b_{ir} + W_{hr}\text{block\_grad}(h_{t-1}) + b_{hr}) \tag{18}$$

$$z_t = \sigma(W_{iz}x_t + b_{iz} + W_{hz}\text{block\_grad}(h_{t-1}) + b_{hz}) \tag{19}$$

$$n_t = \tanh(W_{in}x_t + b_{in} + r_t \star (W_{hn}\text{block\_grad}(h_{t-1}) + b_{hn})) \tag{20}$$

$$h_t = (1 - z_t) \star n_t + z_t \star h_{t-1} \tag{21}$$

In summary we use $h_t$ as the hidden state of the recurrent network, $r_t$, $z_t$ and $n_t$ as the network gates, $\star$ as the term-by-term product, and $W.$ and $b.$ as the weights and bias respectively. One can show that applying e-prop in a classical GRU network is mathematically equivalent to applying BPTT in the network above.

In simulations, we evaluate the performance as the phoneme classification accuracy on the test set. We find that CLAPP-s achieves 61.7% with BPTT and 58.6% with e-prop; but the latter can be implemented with purely local learning rules by relying on eligibility traces [Bellec et al., 2020]. In comparison, phoneme classification from the last feedforward layer before the RNN only yields 52.4% accuracy.

**Biologically plausible computation of the score $u_t^{t+\delta t}$**  We think of the loss $\mathcal{L}_{CLAPP}^t$ in Equation 3 as a surprise signal that is positive if the prediction is wrong, either because a fixation has been wrongly predicted as a saccade or vice-versa. Surprising events are indicated by physiological markers of brain activity such as the EEG or pupil dilation. Moreover, the activity of neuromodulators such as norepinephrine, acetylcholine, and partially also dopamine is correlated with surprising events; an active sub-field of computational neuroscience attempts to relate neuro-modulators to surprise and uncertainty [Angela and Dayan, 2005, Nassar et al., 2012, Ostwald et al., 2012, Schwartenbeck et al., 2013, Heilbron and Meyniel, 2019, Liakoni et al., 2021].

In analogy to the theory of reinforcement learning, where abstract models have been successfully correlated with brain activity and dopamine signals well before the precise brain circuitry necessary to calculate the dopamine signal was known [Dabney et al., 2020], we take the view that surprise signals exist and can be used in the models, even if we have not yet identified a circuit to calculate them. The neuromodulator signal in our model would be 1 if $\mathcal{L}_{CLAPP}^t > 0$ and zero otherwise. Thus the exact value of $\mathcal{L}_{CLAPP}^t$ is not needed.

Nevertheless, let us try to sketch a mechanism to compute this signal. Every neuron $i$ has access to its 'own' internal dendritic signal $\hat{z}_i^t = \sum_j W_{ij}^{pred} c_j^t$ interpretable as the dendritic prediction of somatic activity [Urbanczik and Senn, 2014]. What we need is the product $z_i^{t+\delta t} \hat{z}_i^t$ and then we need to sum over all neurons. Four insights are important. First, a potential problem is that the dendritic prediction $\hat{z}_i^t$ is *different* from the actual activity $z_i^{t+\delta t}$ that is driven in our model by feedforward input. However, the work of Larkum et al. [1999] has shown that neurons emit specific burst-like signals if both dendrite and soma are activated. The product $z_i^{t+\delta t} \hat{z}_i^t$ can be seen as a detector of such coincident events. Second, if bursts indicate such coincident events, then the burst signals of many neurons need to be summed together, which could be done either by an interneuron in the same area (same layer of the model) or by neurons in a deep nucleus located below the cortex. The activity of this nucleus would serve as one of the inputs of the nucleus that actually calculates surprise. Third, following ideas on time-multiplexing in Payeur et al. [2021], the burst signal can be considered as a communication channel that is *separate* from the single-spike communication channel for the feedforward network used for inference. Fourth, as often in neuroscience, positive and negative signals must be treated in different pathways (the standard example is ON and OFF cells in the visual system), before they would be finally combined with the saccade signal $y$ to emit the binary surprise signal $\gamma_t = y^t H^t$ that is broadcasted to the area corresponding to one layer of our network. An empirical test of this suggested circuitry is out of scope for the present paper but will be addressed in future work.

---

[4]https://pytorch.org/docs/stable/generated/torch.nn.GRU.html

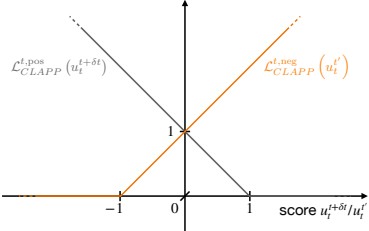

Figure 6: Illustration of the positive ($y = +1$, gray) and negative ($y = -1$, orange) part of the CLAPP loss, see Equation 3

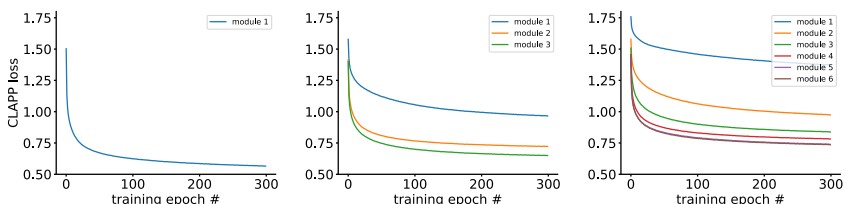

Figure 7: CLAPP-s training losses for encoders split into 1, 3 or 6 gradient-isolated modules.

Table 6: Linear classification test accuracy (%) on STL-10 with features from a VGG-6 encoder trained with CLAPP-s for different sizes of gradient-isolated modules.

| # modules | # layers per module | Test accuracy (%) |
|---|---|---|
| 6, i.e. layer-wise (CLAPP) | 1 | 74.0 |
| 4 modules upper | 3,1,1,1 | 75.4 |
| 4 modules lower | 1,1,1,3 | 76.2 |
| 3 modules | 2 | 77.4 |
| 2 modules | 3 | 77.6 |
| 1 module (end-to-end) (see Table 1) | 6 | 80.3 |

**Preferred patch visualisation for random encoder** As a control, we repeat the preferred patch visualisation analysis, as in Figure 3 a, for the random encoder, i.e. a network with random fixed weights. The result is shown in Figure 8 b, in comparison with the analysis of an encoder trained with CLAPP. For CLAPP, higher layers extract higher-level features creating a hierarchy, whereas for the random encoder, no clear hierarchy is apparent across layers. Together with the non-informative t-SNE embedding of the random encoder (Figure 3 b), this suggest that a convolutional architecture alone does not yield hierarchical representations.

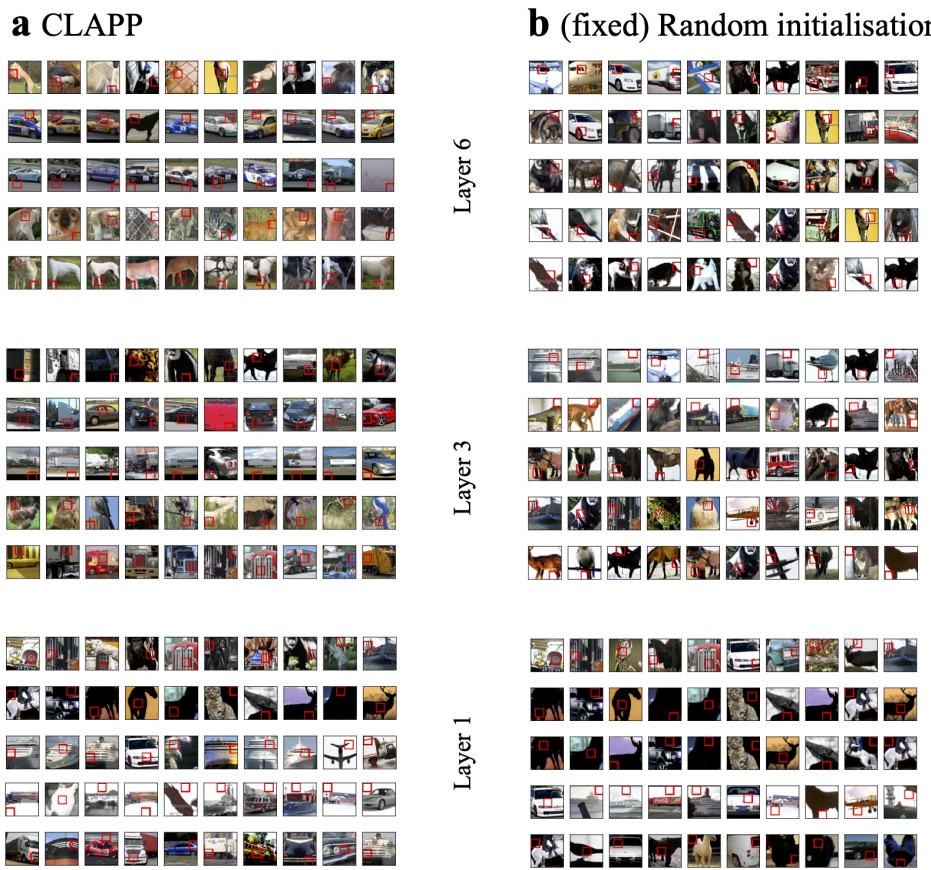

Figure 8: (As Figure 3 a) Red boxes in STL-10 images indicate patches that best activate a specific neuron (rows) in **a** a network trained with CLAPP or **b** a random encoder with random weights fixed at initialisation. For CLAPP, layer 1 extracts simple features like gratings or uniform patches, whereas higher layers extract richer features like object parts. For the random encoder, no clear hierarchy is apparent across layers.

**Gradient flow through MaxPooling layers** In fact, MaxPooling can be viewed as a simple model of lateral inhibition which provides a learning rule compatible with Equation 1, without introducing approximations and without blocking gradients below the MaxPool operator.

The idea is that $2 \times 2$ MaxPooling can be viewed as a simple model of lateral inhibition between the 4 neurons involved. During inference, this inhibition enforces only one of the four neurons to be active. We use the following notation for the output of the MaxPool operator $z'^{t}_{j'} = \max\{z^{t}_{j_0}, z^{t}_{j_1}, z^{t}_{j_2}, z^{t}_{j_3}\}$, where $z^{t}_i$ is defined as in the main paper. We define $c'^{t}$ accordingly for the context layer, if it includes a MaxPool operator.

Then, following the derivation from the main text, the learning rule is proportional to the gradient $\frac{\partial u^{t+\delta t}_t}{\partial W_{ij}}$, but now $u^{t+\delta t}_t$ is defined using the output of the pooling operators: $u^{t+\delta t}_t = \sum_{k',j'} z'^{t+\delta t}_{j'} W^{\text{pred}}_{j'k'} c'^{t}_{k'}$. Since the partial derivative over the MaxPool operator is either 1 (the

neurons is active), or 0 (for the other three neurons, which are inhibited), $\frac{\partial u_t^{t+\delta t}}{\partial W_{ij}}$ is either $\left(\sum_{k'} W_{j'k'}^{\mathrm{pred}} c'_{k'}^t\right) \cdot \sigma'(a_{j'}^{t+\delta t}) x_i^{t+\delta t}$ if $j = j'$ (the neuron is active), or 0 if $j \neq j'$ (the neuron is inhibited). Hence, and without further approximation, the learning rule is only applied if the neuron is active, in which case $\frac{\partial u_t^{t+\delta t}}{\partial W_{ij}}$ takes the form 'dendritic signal $\times$ post $\times$ pre', and the resulting learning rule is compatible with Equation 1.

**Predicting from higher layers**   We ran CLAPP-s with the context representation $c^t$ coming from one layer above the predicted layer $z^{t+\delta t}$ (except for the last layer, where $c^t$ and $z^{t+\delta t}$ come from the same layer). Linear classification performance on STL-10 still grows over layers but only yields 72.4 % test accuracy when classifying from the last layer. In comparison defining $c^t$ to be the same layer as $z^t$ reached 75.0%.