# OpenReview forum: "Local plasticity rules can learn deep representations using self-supervised contrastive predictions"
_NeurIPS.cc/2021/Conference — NeurIPS 2021 Poster_

### Official Review · Reviewer_5YhK · 2021-06-28

**Rating:** 7
**Confidence:** 5

**Summary:**

The authors propose a biologically plausible implementation of contrastive self supervised learning. They solve the problem of remembering negative samples by relying on saccades, taking the post-saccade incoming signal to be a negative sample. They perform a number of comparisons showing the viability of their approach.

--------------------------------------
Post-rebuttal update:
Due to the increased quality of the work stemming from added clarifications and a fuller discussion of biological plausibility, I have increased my score from 5 to 7.

**Limitations And Societal Impact:**

Yes

**Main Review:**

The paper is novel, interesting and well written. In particular I really like the idea of using saccades to solve the problem of memorizing negative samples. In general I think the paper is strong but to be acceptable for submission a number of claims need to be either toned down or further substantiated, see below.

### Strenghts
(in no particular order).
* This is a solid work in a relevant field. In particular, the authors astutely take advantage of signals already available in the visual cortex to circumvent the negative sample problem.
* The experiments are more or less thorough. I would have liked to see more results and error bars but when experiments take 4 days on 4 V100s, there is a limit to what can feasibly be done.
* The ablation study is also more or less convincing of the points the authors want to make.

### Weaknesses
(in the order of appearance in the paper.)
* The authors criticize prior work for the need to wait for convergence (line 34). However, the only reason their model avoids this is a simplifying assumption mentioned in line 77, the assumption that lateral and feedback connections do not affect the activity. In other words, if the authors relax this simplifying assumption (for more realism for example), their dynamics would again be recurrent and they would find themselves in the same situation as prior work.
* Some references for recent work are missing. For instance, there are biologically plausible models that do not have the issues mentioned in lines 101-102 (e.g. https://arxiv.org/abs/2011.15031, although the author's proposed work is more general than this since 2011.15031 only considers shallow networks.)
* Applying the proposed algorithm to auditory pathways seems a stretch (lines 108-109). In this case, as opposed to the auditory pathway where the negative sample can be deduced based on saccades, the presence of a negative sample has to be deduced. The authors assume that this is a solved problem, however, I think this is very non-trivial. Especially since the goal of the proposed circuit itself is to be able to do things like distinguishing different input sources. This can be alleviated by using cross-modality inputs, e.g. visual cues to distinguish sources, but in this case, competing algorithms would also be take advantage of the cross-modality input.
* (line 164) It should be mentioned that the loss in Eq. 3 is only for one layer and each layer has its own loss. That is, there is an implicit stop gradient preventing the weights of the previous layers to accumulate gradients.  This is made more confusing by not having any layer indices, so it looks like there is a single loss objective and a single deicision made regarding whether the saccade/fixation was correctly classified. I think in general the paper would be more clear if the layer indices were present as well.
* Computation of $H^t$ in line 180. The computation of whether or not the saccade was correctly predicted is done according to the definition on line 160. How is this computation actually performed? Suppressing the time sub and superscripts and including the neuron index instead we have $u = \sum_i z_i W_{ij} c_j$. Now  $z_i$ and  $W_{ij} c_j$ are both local to the $i$'th neuron and the product can be computed. But how is this summed? Is there an interneuron that connects to every neuron and sums these? If so, the product  $z_i W_{ij} c_j$ needs to be the output of the neuron. But this is not the case. Altenratively, each neuron can broadcast its own product through some modulatory compound and this can be averaged to give the overall sentiment. This would be plausible but not biologically realistic (also is not what is actually implemented). Unless I misunderstood what is happening, I think this is the biggest weakness of the paper and needs to be fully addressed.
* Line 188: 'We show in appendix C that ... '. This sounds like there is something akin to an analytic demonstration if not a proof, therefore is sounds a lot stronger than what was actually claimed at the end of appendix C which is that empirically it seems fine.
* Line 208, 'we allow gradient flow through max-pool' This statement should be expanded (perhaps in the appendix). I cannot tell based on the paper if this is cause for worry or just a trivial assumption.
* Line 219, 'comparable results were obtained', please provide the resulting figures in the appendix and the details of the implementation in the code. It seems like it would not be a totally trivial change in implementation. It would be good for the authors to clarify these in case anyone would want to reproduce their results in the future.
* Lines 230-234, emergence of a hierarchy. This is a weak part of the empirical demonstrations. The authors show that the neurons of later layers respond to more complex shapes. Is this not expected of a conv-net? Afterall the receptive field of the first layer is just 3x3, it cannot possibly see anything more complex. Similarly, deeper layers would have wider receptive fields. I think the definition of hierarchical representation - that is deeper representations are built out of the smaller representations - is in fact built into a layered-conv net and therefore the hierarchical claims are all trivial. If the authors meant a useful or interpretable hierarchical representation, then this was addressed elsewhere in the paper.
* Lines 236-246 Straight line in t-SNE. I am baffled by this. t-SNE is a non-linear mapping and has many bells and whistles that can change the final shape of the resulting representation. As such I'm not sure if this straight line business means anything. It seems like the 3rd layer can do the vehicles/animals discrimination with a slightly bent line so they are similar to me. The following paragraph using a linear decoder for classification is a much better way of seeing performance.



**Time Spent Reviewing:**

6

---

> ### Author Response · Authors · 2021-08-09
> **Author response to Reviewer 5YhK (Part 1)**
>
> We thank the reviewer for the thorough review and excellent comments. We appreciate the overall positive reaction and the constructive feedback, which we will address in detail below:
>
> 1. **Convergence** We appreciate bringing up the question of recurrence, however, we disagree that after 'relaxing this simplifying assumption' we would 'find ourselves in the same situation as' equilibrium propagation or contrastive divergence.
> We provide below four reasons why we think the recurrence is different in our model. We realize that the formulation 'For simplicity ... ' in line 77 is misleading and we will replace it by a new paragraph in the final version: (i) Algorithmically, using the recurrent dendritic input only for plasticity and not for inference is *not* a simplification but it is meant to implement contrastive predictive coding, as intended for CLAPP. Thus our algorithm operates with zero (or one, depends on the counting) recurrent iteration. This is different from equilibrium propagation and contrastive divergence which operate on fully converged representations (hundreds of time steps are necessary with equilibrium propagation in CNNs [Laborieux et al., 2021]). Empirically, it was found by several labs that the update rule of equilibrium propagation does not work properly if applied 'early', i.e., before complete convergence.
> (ii) We do apply CLAPP with RNNs on audio inputs. Hence, recurrent activity influencing inference is not excluded, and it works even if the input changes at every time step (which is not possible with equilibrium based learning rules because they require a stationary input).
> (iii) In equilibrium propagation and contrastive divergence, the convergence propagates information about the label to the lower layers.
> In the setting where $c_t^l = z_t^l$ we have build deep representations without any downward signalling of information.
> (iv) There is neuroscientific and theoretical insight showing the dentritic inputs to basal and apical dendrites influence activity and plasticity in very different ways [Larkum et al., 1999, Dudman et al., 2007,Major et al., 2013, Urbanczik and Senn, 2014, Payeur et al., 2021] and the community has not yet converged to a quantitative model which would model that with precision.
> In general, we do not exclude the possibility that the dendritic input may influence neural activity later on (after inference), but we leave this to future work.
>
> 2. We thank the reviewer for the reference **Golkar et al., [2020]**, which we will include in the related work section, along with literature on difference target propagation.
>
> 3. **Auditory pathway** Contrastive predictive coding was originally designed for time-series, such as audio, and it seems natural to us to apply CLAPP to audio data to model learning of the auditory pathway. However, we fully agree that the notion of 'saccade', that we exploit in the visual system, does not make sense in the auditory system. Hence, we consider the application of CLAPP to auditory signals as benchmark application, and not as a neuroscientifically relevant study. We will stress this point in the final version.
> Nevertheless we may speculate on where a broadcast signal indicating negative samples could arise in the auditory system.  We see three possible candidates: (i) cross-modal input (indicating a change in head or gaze direction); (ii) change in attentional focus (as already mentioned in the paper),  and (iii)  signal/speaker-identity inferred from blind source separation, which can be done on low-level representation with biologically plausible learning rules, see e.g. Hyvärinen and Oja  [1997]. Points (i) - (iii) could be combined such that attentional focus switches to a different source after a change in head direction which generates a switch-signal that is communicated to all layers of the auditory processing network.
>
> 4. **Layer indices** Thanks for the comment, we totally agree and will add comments on the layer-wise loss, stop-gradients and will add layer-indices in the final version.
>
> 5. **Computation of H/u** We agree that this is an important point. Indeed, plasticity is active if and only if $u < 1$ during fixation ($y=1$), and $u > -1$ after a saccade ($y=-1$). Some circuitry is needed to evaluate $u$ and we have not discussed this so far. We suggest to add a paragraph in the discussion of the updated manuscript to address this point.
> Let us try to sketch a mechanism to compute this signal: As mentioned by the reviewer, every neuron $i$ has access to its 'own' internal dendritic signal $\hat{z}_i^t = \sum_j W^{pred,ij} c^t_j$ interpretable as the dendritic prediction of somatic activity [Urbanczik et al 2014].
> What we need is the product $z_i^{t+\delta t} \hat{z_i}^t$ and then we need to sum over all neurons. Four insights are important.
> First, and as mentioned by the reviewer, a potential problem is that the dendritic prediction $\hat{z}_i^t$ is *different* from the actual activity $z_i^{t+\delta t}$ that is driven in our model by feedforward input. However, the work of Mathew Larkum and others has shown that neurons emit specific burst-like signals if both dendrite and soma are activated [Larkum 1999]. The product $z_i^{t+\delta t} \hat{z_i}^t$ can be seen as a detector of such coincident events.  Second, if bursts indicate such coincident events, then the burst signals of many neurons need to be summed together, which could be done either by an interneuron in the same area (same layer of the model) or by neurons in a deep nucleus located below the cortex. The activity of this nuclues would serve as one of the input of the nucleus that actually calculates surprise.
> Third, following ideas of Payeur et al [2021] the burst signal can be considered as a communication channel that is *separate* from the single-spike communication channel for the feedforward network used for inference.
> Fourth, as often in neuroscience, positive and negative signals must be treated in different pathways (the standard example is ON and OFF cells in the visual system), before they would be finally combined with the saccade signal $y$ to emit  the binary surprise signal $\gamma_t=y^t H^t$ that is broadcasted to the area corresponding to one layer of our network.
> An empirical test of this suggested circuitry is out of scope for the present paper but will be addressed in future work.
> We think of the loss $L$, which is a function of $u\cdot y$, as a surprise signal that is positive if the prediction is wrong, either because a fixation has been wrongly predicted as a saccade or vice-versa. Surprising events are indicated by physiological markers of brain activity such as the EEG or pupil dilation; moreover, the activity of neuromodulators such as norepinephrine, acetylcholine, and partially also dopamine is correlated with surprising events; an active sub-field of computational neuroscience attempts to relate neuro-modulators to surprise and uncertainty [Angela and Dayan, 2005, Nassar et al., 2012, Schwartenbeck et al., 2013, Liakoni et al., 2021]. In analogy to the  theory of reinforcement learning where abstract models have been successfully correlated with brain activity and  dopamine signals well before the precise brain circuitry necessary to  calculate the dopamine signal was known, we take the view that surprise signals exist and can be used in the models, even if we have not yet identified a circuit to calculate them.
>
> **Author responses to remaining points in next comment**
>
> References:
> - J.  Y.  Angela  and  P.  Dayan.   Uncertainty,  neuromodulation,  and  attention.Neuron,  46(4):681–692,2005
> - W.  Dabney,   Z.  Kurth-Nelson,   N.  Uchida,   C.  K.  Starkweather,   D.  Hassabis,   R.  Munos,   andM. Botvinick.  A distributional code for value in dopamine-based reinforcement learning.Nature,577(7792):671–675, 2020.
> - J.  T.  Dudman,  D.  Tsay,  and  S.  A.  Siegelbaum.   A  Role  for  Synaptic  Inputs  at  Distal  Dendrites:Instructive Signals for Hippocampal Long-Term Plasticity.Neuron, 56(5):866–879, dec 2007.  ISSN08966273.  doi:  10.1016/j.neuron.2007.10.020
> - A. Hyvärinen and E. Oja.  A fast fixed-point algorithm for independent component analysis.Neuralcomputation, 9(7):1483–1492, 1997.
> - A. Laborieux,  M. Ernoult,  B. Scellier,  Y. Bengio,  J. Grollier,  and D. Querlioz.  Scaling equilibriumpropagation to deep convnets by drastically reducing its gradient estimator bias.Frontiers in neu-roscience, 15:129, 2021
> - M. E. Larkum, J. J. Zhu, and B. Sakmann.  A new cellular mechanism for coupling inputs arriving atdifferent cortical layers.Nature, 398(6725):338–341, 1999.  ISSN 00280836.  doi:  10.1038/18686.
> - V. Liakoni, A. Modirshanechi, W. Gerstner, and J. Brea.  Learning in volatile environments with thebayes factor surprise.Neural Computation, 33(2):269–340, 2021
> - G. Major, M. E. Larkum, and J. Schiller. Active properties of neocortical pyramidal neuron dendrites.Annual  review  of  neuroscience,  36:1–24,  jul  2013.   ISSN  1545-4126.   doi:  10.1146/annurev-neuro-062111-150343
> - M. R. Nassar, K. M. Rumsey, R. C. Wilson, K. Parikh, B. Heasly, and J. I. Gold. Rational regulationof learning dynamics by pupil-linked arousal systems.Nature neuroscience, 15(7):1040–1046, 2012.
> - A. Payeur, J. Guerguiev, F. Zenke, B. A. Richards, and R. Naud. Burst-dependent synaptic plasticitycan coordinate learning in hierarchical circuits.Nature neuroscience, pages 1–10, 2021.
> - R. Urbanczik and W. Senn.  Learning by the dendritic prediction of somatic spiking.Neuron, 81(3):521–528, 2014.
> - P. Schwartenbeck, T. FitzGerald, R. Dolan, and K. Friston.  Exploration, novelty, surprise, and freeenergy minimization.Frontiers in psychology, 4:710, 2013
> - D. L. Yamins and J. J. DiCarlo. Using goal-driven deep learning models to understand sensory cortex.Nature neuroscience, 19(3):356–365, 2016.

---

> > ### Comment · Reviewer_5YhK · 2021-08-11
> > **Post-rebuttal comment**
> >
> > I thank the authors for their detailed response. In general I think the authors address most of my concerns. Here are two points that remain unclear to me.
> >
> > **5. Computation of $H,u$.** I agree with the response of the authors. In short, to make their proposed algorithm work, some kind of multiplexing is required. The authors suggest burst encoding as a biologically motivated method of multiplexing. This is all great. However, since 1. the computation of $u$ is integral to the algorithm and 2. multiplexing as a whole is still a bit controversial, I would strongly suggest that the authors do not bury this part of their network in a paragraph in the discussion section, but instead treat it openly and on equal footing with the other parts of their biological implementation and mention it in Sec. 2.
> >
> > Furthermore, the authors prominently mention in the abstract and introduction that their network does not need to back-propagate error signals. However, as has become apparent, they do need to use multiplexing which is common in networks that do perform backpropagation. So again, in the spirit of honesty, I would suggest that this becomes clear to the reader in the same vein, either in the abstract or the introduction.
> >
> > **8. Emergence of hierarchy.** I am not convinced by the arguments. Rephrasing my concern from the review: *Any representation that comes out of a convolutional network, which is 1. useful and 2. knows about patterns larger than the kernel size used, will be hierarchical.* I don't really see how there is a way around this. In a convolutional network, if the representation knows about large scale structure in the sample, by definition it has to be built out of smaller scale structures. I am a also confused by the reference to Fig. 3b. What I get out of the ablation study (using untrained conv net) is that the untrained conv net just does not have a useful representation.
> >
> > Perhaps a more pertinent question here is, what do the authors define as a hierarchical representation? To me, a hierarchical visual representation is one that starts by considering local neighborhoods, and each successive processing of the data builds features that are larger scale and are comprised of the smaller parts discovered previously. So this begs three questions: 1. Do the authors have a different notion of what a hierarchical visual representation means? 2. If they subscribe to the same notion of hierarchical visual representation, is this not by definition what a convolutional network does? 3. How does Fig 3.b (that the authors use to ablate the effect of convolutions), which considers separability of classes vs. layer number, relate to this notion of hierarchy? Even in non-hierarchical multi-layer networks, the linear separability of different classes increases as one gets closer to the final layer.

---

> > > ### Author Response · Authors · 2021-08-12
> > > **Author response to the post-rebuttal comments of Reviewer 5YhK**
> > >
> > > We highly appreciate the reviewer's comments and are happy to provide more clarification.
> > >
> > > - **Computation of $H$ and $u$** We appreciate the positive comment. We agree with the reviewer, that multiplexing is sill debated in the field and we will mention the need for multiplexing in the introduction.  Furthermore, we will add the new paragraph (see previous reply), explaining the computation of $H$ and $u$, in Section 2 (instead of the Discussion Section) and we will also discuss there the relation of multiplexing in our model with that in current models of biologically plausible backpropagation.
> > >
> > > - **Emergence of hierarchy** We implicitly adopted a common notion in the field , but we realise now that our paper misses a clear definition of the concept 'hierarchical'. Following, amongst others, [Fukushima,  1988,  Riesenhuber and Poggio, 1999, LeCun, 2012, Lillicrap et al., 2020, Payeur et al., 2021]  we will define hierarchical representation  in the final version of the text as follows: "A (layered) hierarchy of features (i) builds higher-level features out of lower-level ones, and (ii) provides more useful features in higher layers, where usefulness will be measured by classification with a linear classifier."  A CNN with fixed random weights and with standard weight sharing, by definition, constructs higher features out of lower ones, but fails to meet condition (ii) and is thus not hierarchical according to this definition. We apologize that the vague notion of 'hierarchical' has led to some misunderstandings. Given the above definition of hierarchy, we argue that our analyses in Figure 3 and 4 are legitimate ways to test for hierarchical representations. We will add this definition and the references in the updated version.
> > >
> > > References:
> > >
> > > - K.  Fukushima.   Neocognitron:  A  hierarchical  neural  network  capable  of  visual  pattern  recognition.Neural Networks, 1(2):119–130, 1988.  ISSN 08936080.  doi:  10.1016/0893-6080(88)90014-7
> > > - Y. LeCun.  Learning invariant feature hierarchies.  InEuropean conference on computer vision, pages496–505. Springer, 2012
> > > - T. P. Lillicrap, A. Santoro, L. Marris, C. J. Akerman, and G. Hinton. Backpropagation and the brain.Nature Reviews Neuroscience, 21(6):335–346, 2020.
> > > - A. Payeur, J. Guerguiev, F. Zenke, B. A. Richards, and R. Naud. Burst-dependent synaptic plasticitycan coordinate learning in hierarchical circuits.Nature neuroscience, pages 1–10, 2021
> > > - M.   Riesenhuber   and   T.   Poggio.Hierarchical   models   of   object   recognition   in   cortex.Nat.   Neurosci.,    2(11):1019–25,    1999.ISSN   1097-6256.doi:10.1038/14819.URLhttp://www.ncbi.nlm.nih.gov/pubmed/10526343.

---

> > > > ### Comment · Reviewer_5YhK · 2021-08-12
> > > > **Comment on Author response to the post-rebuttal comments of Reviewer 5YhK**
> > > >
> > > > Thank you for the clarification.

---

> ### Author Response · Authors · 2021-08-09
> **Author response to Reviewer 5YhK (Part 2)**
>
> 6. **Apendix C** We understand the concern and will change the wording to: 'In appendix C, we show empirically that...'.
>
> 7. **MaxPooling layers** Thanks for pointing this out.
> In fact, it is possible to provide a more precise statement because Max-pooling can be viewed as a simple model of lateral inhibition and our theory provides a learning rule compatible with the equation prototype of equation (1) without introducing approximations and without blocking gradients below the max-pooling operator.
> We suggest to expand this comment in the main text and to show this in a new paragraph with a derivation in the appendix. We sketch the derivation below:
> The idea is that $2 \times 2$ Max-pooling can be viewed as a simple model of lateral inhibition between the 4 neurons involved. During inference, it models that only one of the four neurons can be active simultaneously. We use the following notations so that the output of the max-pooling operator is denoted as ${z'}_{j'}^t = \max \lbrace {z}\_{j\_0}^t,  {z}\_{j_1}^t, {z}\_{j_2}^t, {z}\_{j_3}^t \rbrace$ where ${z}\_{i}^t$ is defined as in the main paper.
> We also define ${c'}^{t}$ similarly for the context layer if it includes a max-pooling operator.
> Then, following the derivation from the main text, the learning rule is proportional to the gradient $\frac{\partial u}{\partial W\_{ij}}$
> but now $u$ is defined using the output of the pooling operators:
> $u = \sum\_{k',j'}  {z'}^{t+\delta t}\_{j'} W\_{j'k'}^{\mathrm{pred}} {c'}^{t}\_{k'}$.
> Since the partial derivative over the max-pooling operator is either $1$ if the neuron is active and $0$ for the other three neurons which are inhibited, $\frac{\partial u}{\partial W\_{ij}}$ is either $0$ if $j \neq j'$ (the neuron is inhibited) or $\left(\sum\_{k'} W^{\mathrm{pred}}\_{j'k'} {c'}^{t}\_{k'}\right) \cdot \sigma'(a_j^{t+\delta t}) x_i^{t+\delta t}$ if $j = j'$ (the neuron is active).
> Hence, without further approximation, the learning rule is only applied if the neuron is active, in which case $\frac{\partial u}{\partial W\_{ij}}$ takes the form 'dendritic signal $\times$ post $\times$ pre' and the resulting learning rule is compatible with the plausible learning rule prototype from equation (1).
>
> 8. **Line 219, 'comparable results were obtained'** We ran CLAPP-s with the context representation $c^t$ coming from one layer above the predicted layer $z^{t+\delta t}$ (except for the last layer, where $c^t$ and $z^{t+\delta t}$ come from the same layer) and achieved 72.4 \% test accuracy on STL-10 when classifying from the last layer.
> In comparison defining $c^t$ to be the same layer as $z^t$ reached 75.0\%. In the final version, we will replace the vague 'comparable results' phrase by the precise number. In the code, this modification is implemented as a keyword option for easy reproducibility.
>
> 9. **Lines 230-234, emergence of a hierarchy** We fully understand this concern, but hope to convince the reviewer that a convolutional architecture *alone* is *not* capable of producing hierarchical representations. To this end, our paper features multiple controls with a randomly initialized convolutional encoder and we show that such random representations are *non-hierarchical*, as can be seen by the unstructured tSNE clustering in Fig. 3B (same result for a wide range of tSNE parameters) and the bad linear classification performance in Fig 4. More specifically about the selectivity analysis mentioned by the reviewer, we will add a preferred patch analysis for the random encoder, as in Fig 3a, in the appendix of the final version, and comment on our findings with a convolutional network with random filters in the main text.
>
> 10. **Lines 236-246 Straight line in t-SNE** We agree that t-SNE embeddings should not be over-interpreted and will adapt the figure (removing straight line) and caption to avoid over-interpretation of the t-SNE visualisation. In our experiments (over a wide range of t-SNE parameters), multiple runs of t-SNE gave qualitatively similar results e.g. birds always being next to planes at layer 6 (except for random orientation and ordering changes expected from a stochastic method). We thus believe that t-SNE is still valid as a *qualitative visualisation* tool but perfectly agree that the linear classification is a much better metric.
>
> References:
> - J.  Y.  Angela  and  P.  Dayan.   Uncertainty,  neuromodulation,  and  attention.Neuron,  46(4):681–692,2005
> - W.  Dabney,   Z.  Kurth-Nelson,   N.  Uchida,   C.  K.  Starkweather,   D.  Hassabis,   R.  Munos,   and M. Botvinick.  A distributional code for value in dopamine-based reinforcement learning.Nature,577(7792):671–675, 2020.
> - J.  T.  Dudman,  D.  Tsay,  and  S.  A.  Siegelbaum.   A  Role  for  Synaptic  Inputs  at  Distal  Dendrites:Instructive Signals for Hippocampal Long-Term Plasticity.Neuron, 56(5):866–879, dec 2007.  ISSN08966273.  doi:  10.1016/j.neuron.2007.10.020
> - A. Hyvärinen and E. Oja.  A fast fixed-point algorithm for independent component analysis.Neuralcomputation, 9(7):1483–1492, 1997.
> - A. Laborieux,  M. Ernoult,  B. Scellier,  Y. Bengio,  J. Grollier,  and D. Querlioz.  Scaling equilibriumpropagation to deep convnets by drastically reducing its gradient estimator bias.Frontiers in neu-roscience, 15:129, 2021
> - M. E. Larkum, J. J. Zhu, and B. Sakmann.  A new cellular mechanism for coupling inputs arriving atdifferent cortical layers.Nature, 398(6725):338–341, 1999.  ISSN 00280836.  doi:  10.1038/18686.
> - V. Liakoni, A. Modirshanechi, W. Gerstner, and J. Brea.  Learning in volatile environments with thebayes factor surprise.Neural Computation, 33(2):269–340, 2021
> - G. Major, M. E. Larkum, and J. Schiller. Active properties of neocortical pyramidal neuron dendrites.Annual  review  of  neuroscience,  36:1–24,  jul  2013.   ISSN  1545-4126.   doi:  10.1146/annurev-neuro-062111-150343
> - M. R. Nassar, K. M. Rumsey, R. C. Wilson, K. Parikh, B. Heasly, and J. I. Gold. Rational regulationof learning dynamics by pupil-linked arousal systems.Nature neuroscience, 15(7):1040–1046, 2012.
> - A. Payeur, J. Guerguiev, F. Zenke, B. A. Richards, and R. Naud. Burst-dependent synaptic plasticitycan coordinate learning in hierarchical circuits.Nature neuroscience, pages 1–10, 2021.
> - R. Urbanczik and W. Senn.  Learning by the dendritic prediction of somatic spiking.Neuron, 81(3):521–528, 2014.
> - P. Schwartenbeck, T. FitzGerald, R. Dolan, and K. Friston.  Exploration, novelty, surprise, and freeenergy minimization.Frontiers in psychology, 4:710, 2013
> - D. L. Yamins and J. J. DiCarlo. Using goal-driven deep learning models to understand sensory cortex.Nature neuroscience, 19(3):356–365, 2016.

---

### Official Review · Reviewer_hGsk · 2021-07-16

**Rating:** 8
**Confidence:** 4

**Summary:**

This paper proposes a new Hebbian-like learning algorithm for neural networks that is able to produce hierarchical representations. This new self-supervised learning rule, which is based on the ideas of Contrastive Predictive Coding (CPC), obeys several key biological constraints (such as locality) while attaining similar benchmarks on several standard ML-classification tasks as previous, unconstrained rules.

**Ethical Concerns:**

none.

**Limitations And Societal Impact:**

The authors have acknowledged that the current learning rule in its current form is data- and compute-intensive. However, the locality in space and time allows it to be used in neuromorphic hardware, potentially reducing energy consumption significantly.

**Main Review:**

The gap between biologically plausible learning algorithms and implausible once such as backpropagation has recently attracted a lot of interest. The authors' approach here is quite original, as they build their learning rule around CPC, and, quite interestingly, assume that the brain self-generates contrastive samples by e.g. performing saccades. That's a great idea! (I hadnt seen that voiced before.) The paper is well written and the exposition is quite scholarly, in that the authors clearly delineate how their model compares to other methods in the field, and how the different aspects can be traced back to other studies (locality of rule, the avoidance of BP, the presence of modulating factors, etc.). The final simulations are convincing, and show that the rule does generate hierarchical and meaningful representations.
I only have a few suggestions/questions:

(1) Why does CLAPP require many more negative samples on video than on STL-10? That wasnt clear to me.

(2) The distinction between the predicted layer z and predicting layer c gets confusing at times. On the one hand, Figure 2c and L75-77 seem to indicate that the feedforward network of z’s is the only network needed (Note specially L85-94). On the other hand, L170-174 and the remaining equations seem to indicate that z and c are separate neurons with separate feedforward connections. The clarity of this distinction could be improved both in text and figures.

(3) L36-37: “The present paper demonstrates that deep representations can emerge from a local, biologically plausible and unsupervised learning rule”. Despite having some components which are biologically plausible, the author introduce a shortcut which seems to violate biology: the learning rule requires a synaptic connection $W^{pred}$ that does not affect the neuron's firing but only synaptic plasticity. In other words, how could a signal that arrives at an apical dendrite influence synaptic plasticity at a basal dendrite while not directly influencing the activity of the neuron itself? Please clarify or word more carefully.



**Time Spent Reviewing:**

10h (together with a postdoc in the lab)

---

> ### Author Response · Authors · 2021-08-09
> **Author response to Reviewer hGsk**
>
> We thank the reviewer for the thorough review and excellent comments.  We appreciate the overall positive reaction and the constructive feedback, which we will address in detail below:
>
> 1. **Number of negative samples** In the final version we will clarify this in the text. For the video task, we use the batch version of CLAPP, called CLAPP-s which uses $N$ video samples in parallel. Our intuition is that the video dataset is more complex than STL-10 and thus requires globally 'more comparisons (pairs of positive/negative samples)' to converge.
> Since the CLAPP algorithm *linearly* combines updates stemming from multiple negative and positive samples, we expect the same performance as reported in the paper if we run the online CLAPP algorithm for a  longer time (and ADAM implicitly readjusts the effective learning rate); indeed, for the static images we found this expected equivalence. Since the wording `number of negative examples' is misleading, we will change the formulation in the final version of the text.
>
>
> 2.  **Distinction z and c** We thank the reviewer for this valuable comment; this needs to be clarified. Both $z$ and $ c $ are *always* taken from the main feedforward network. They can refer to the same layer ($z^{t,l} = c^{t,l}$) or different layers (concretely we tested the alternative where $z^{t,l+1} = c^{t,l}$), but the algorithm can be implemented in principle with other layer combinations. We will add a clarification on the distinction between z and c at the beginning of section 3.
>
>
> 3. **Role of recurrent weights** Thanks for bringing up this subtle point. This was done deliberatly to simplify the mathematical model, but we do not intend to claim that dendritic activity never influences neuronal (somatic) activity. There is, however, neuroscientific evidence  [Larkum et al., 1999, Dudman et al., 2007, Major et al., 2013, Urbanczik and Senn, 2014] that the inputs to basal and apical dendrites affect the neural activity and the plasticity in different ways.
> Towards a more precise model, where recurrent activity also influences the network activity, note that we already described how to apply our learning rule with a recurrent network, where both types of recurrent connections coexist. In general, we do not rule out influence of dendritic activity on somatic activity in later phases of cortical processing. In the final version, we will elaborate on these considerations and literature in the mentioned paragraph and reword more carefully.
>
>
> References:
> - M. E. Larkum, J. J. Zhu, and B. Sakmann.  A new cellular mechanism for coupling inputs arriving atdifferent cortical layers.Nature, 398(6725):338–341, 1999.  ISSN 00280836.  doi:  10.1038/18686.
>
> - J.  T.  Dudman,  D.  Tsay,  and  S.  A.  Siegelbaum.   A  Role  for  Synaptic  Inputs  at  Distal  Dendrites:Instructive Signals for Hippocampal Long-Term Plasticity.Neuron, 56(5):866–879, dec 2007.  ISSN08966273.  doi:  10.1016/j.neuron.2007.10.020
>
> - G. Major, M. E. Larkum, and J. Schiller. Active properties of neocortical pyramidal neuron dendrites.Annual  review  of  neuroscience,  36:1–24,  jul  2013.   ISSN  1545-4126.   doi:  10.1146/annurev-neuro-062111-150343.
>
> - R. Urbanczik and W. Senn.  Learning by the dendritic prediction of somatic spiking.Neuron, 81(3):521–528, 2014

---

### Official Review · Reviewer_5VGo · 2021-07-18

**Rating:** 6
**Confidence:** 4

**Summary:**

This paper proposes a "Contrastive, Local And Predictive Plasticity" learning rule (CLAPP) that could be implemented by deep neural networks of the brain. The authors claim that it results in deep hierarchical representations akin to those recorded in the brain / from deep image classifiers trained with Backprop, while avoiding the need for a backward pass.

CLAPP relies on a global modulation signal which is related to eye movements: for a relatively static eye, the visual input and its encoding are expected to change little ("positive" samples); whereas every saccade leads to a large change in the image / code ("negative" examples). Another modulating factor is delivered via apical dendrites, which control the learning, but play no role in the feedforward signal transmission (see below for more detail).

Finally, the authors show the CLAPP training on the 6-layer VGG network for images (STL-10 dataset), as well as speech and video. They provide arguments for why they believe their network develops "deep representations" using visualisation (patches that best activate a few  neurons and t-SNE across layers) and probes (linear classifiers trained for each layer separately).

The authors state that the error bars and a proper statistical evaluation are missing due to computational and time constraints.


**More detail:**
The predictive signal computed by the apical dendrites has no impact on the neural firing rate, but it has to be somehow read out across the entire layer  to form, together with the neural outputs, the global prediction of the saccade $u=\sum_j \text{neuron output}_j \times \text{dendritic prediction}_j.$
It is then compared to the presence of saccade signal and used to train:
- the feedforward weights of the given layer,
- the predictive weights (requiring the neuron's action potentials to affect the apical dendrite synapses).

The same signal (gathered from the layer's dendrites), affects the learning of the "prediction source" layer, which applies similar rules, but reversed in time: Instead of updating based on past input and current output, the synapses are updated based on the current input and past output. These result in symmetrical rules for updates of:
- the feedforward weights of the "prediction source" layer,
- the retrograde weights which feed back the layer L's output to the "prediction layer".

**Limitations And Societal Impact:**

The authors state their model is more biologically plausible than the existing proposals, but they do not discuss the new set of assumptions that is required for their model to work. Most importantly, the "broad modulation signal" affecting learning of a given layer needs to be computed based on the predictive dendritic and output signals of the entire layer. The output of the zWc computation needs to be compared to the  "saccade" signal and broadcast back to all neurons (in both the feedforward and the prediction layers), to affect learning in their apical and basal dendrites. In the current implementation, this entire computation happens instantaneously.

**Main Review:**

- This work is original and it clearly states the difference to other approaches of developing biologically plausible learning algorithms.

- There are no new theoretical results, equations are correct and easy to check with basic linear algebra tools.
Any claims about deep hierarchical representations are very hard to evaluate.

    This reviewer is not convinced by the argumentation referring to the tSNE representation (tSNE is an algorithm that fits local neighbourhood, thus a line has no meaning in this image; another run of tSNE or a slight change of parameters could have brought the "birds" class closer to the "furry animals"; there are plenty of samples on that side of the line anyway). The linear decodability is a more useful metric, but it's hard to tell how much of the benefit is due to the increasing receptive field sizes; in the end, the local learning rule (greedy supervised) shows a similar pattern across depth.

- The submission is clearly written and well organised. (But note LL 130--131.)

- This submission is an interesting voice in the ongoing discussion of what are the learning mechanisms used by the neural networks of the brain.








**Time Spent Reviewing:**

4

---

> ### Author Response · Authors · 2021-08-09
> **Author response to Reviewer 5VGo**
>
> We thank the reviewer for the thorough review and excellent comments. We appreciate the overall positive reaction and the constructive feedback, which we will address in detail below:
>
> 1. **Computation of the global prediction u** We agree that this is an important point. Indeed, plasticity is active if and only if $u < 1$ during fixation ($y=1$), and $u > -1$ after a saccade ($y=-1$). Some circuitry is needed to evaluate $u$ and we have not discussed this so far. We suggest to add a paragraph in the discussion of the updated manuscript to address this point.
> We think of the loss $L$, which is a function of $u\cdot y$, as a surprise signal that is positive if the prediction is wrong, either because a fixation has been wrongly predicted as a saccade or vice-versa.  Surprising events are indicated by physiological markers of brain activity such as the EEG or pupil dilation; moreover, the activity of neuromodulators such as norepinephrine, acetylcholine, and partially also dopamine is correlated with surprising events; an active sub-field of computational neuroscience attempts to relate neuro-modulators to surprise and uncertainty [Angela andDayan, 2005, Nassar et al., 2012, Ostwald et al., 2012, Schwartenbeck et al., 2013, Heilbron andMeyniel, 2019, Liakoni et al., 2021]
> In analogy to the  theory of reinforcement learning, where abstract models have been successfully correlated with brain activity and dopamine signals well before the precise brain circuitry necessary to calculate the dopamine signal was known, we take the view that surprise signals exist and can be used in the models, even if we have not yet identified a circuit to calculate them. The neuromodulator signal in our model would be 1 if $L>0$ and zero otherwise. Thus the exact value of $L$ is not needed.
> Nevertheless, let us try to sketch a mechanism to compute this signal. Every neuron $i$ has access to its 'own' internal dendritic signal
> $\hat{z}_i^t = \sum_j W^{pred,ij} c^t_j$
> interpretable as the dendritic prediction of somatic activity [Urbanczik et al., 2014].
> What we need is the product $z_i^{t+\delta t} \, \hat{z_i}^t$ and then we need to sum over all neurons. Four insights are important.
> First, a potential problem is that the dendritic prediction $\hat{z}_i^t$ is *different* from the actual activity $z_i^{t+\delta t}$ that is driven in our model by feedforward input. However, the work of Mathew Larkum  and others has shown that neurons emit specific burst-like signals if both dendrite and soma are activated [Larkum et al., 1999]. The product $z_i^{t+\delta t} \, \hat{z_i}^t$ can be seen as a detector of such coincident events.  Second, if bursts indicate such coincident events, then the burst signals of many neurons need to be summed together, which could be done either by an interneuron in the same area (same layer of the model) or by neurons in a deep nucleus located below the cortex. The activity of this nucleus would serve as one of the inputs of the nucleus that actually calculates surprise.
> Third, following ideas of Payeur et al, [2021], the burst signal can be considered as a communication channel that is *separate* from the single-spike communication channel for the feedforward network used for inference.
> Fourth, as often in neuroscience, positive and negative signals must be treated in different pathways (the standard example is ON and OFF cells in the visual system), before they would be finally combined with the saccade signal $y$ to emit  the binary surprise signal $\gamma_t=y^t H^t$ that is broadcasted to the  area corresponding to one layer of our network.
> An empirical test of this suggested circuitry is out of scope for the present paper but will be addressed in future work.
>
> 2. **This work is original and it clearly states the difference to other approaches of developing biologically plausible learning algorithms** Thanks for this positive remark.
>
> 3. **Theoretical results** We agree that our derivations (e.g., gradient calculation and interpretation)  are not necessarily new 'theory'. If the reviewer prefers, we suggest to rename Section 3 to 'Derivation of a local plasticity rule'. We agree that the evaluation of deep hierarchical representations is hard, but the work of DiCarlo and others (e.g. Yamins et al, [2016]) has established the quantitative read-out by a linear classifier as an accepted  tool in the field.
>
> 4. **Concerning tSNE and evaluation** Indeed, tSNE embeddings are a visualization tool that should not be over-interpreted. We will adapt the figure (removing the straight line) and caption as well as the main text accordingly to avoid any over-interpretation of the tSNE visualisation. In our experiments, multiple runs of tSNE (over a wide range of t-SNE parameters) gave qualitatively similar results e.g. birds always being next to planes at layer 6 (except for random orientation and ordering changes expected from a stochastic method). We thus believe that tSNE is a valid  *qualitative visualisation* tool but agree that the linear classification is a much better quantitative metric. Regarding increasing receptive field sizes: this is a valid concern but our results with fixed random weights indicate that the convolutional feedforward architecture alone is *not* capable of producing a hierarchical representation (see Fig. 3B, right). To highlight this point, we will add in the appendix for comparison a preferred patch analysis as in Fig 3a but for the network with random weights. Greedy supervised training, another candidate of a local rule,  also produces a qualitative hierarchy but (i) it is a supervised algorithm that needs labels and (ii) on STL10 the final performance is 7 percent points below that of CLAPP (66 versus 73 percent). We will change and expand the information in the appendix accordingly.
>
> 5. **The submission is clearly written and well organised. (But note LL 130--131.)** We will improve clarity of LL. 130-131.
>
> 6. **This submission is an interesting voice in the ongoing discussion of what are the learning mechanisms used by the neural networks of the brain** Thanks for the overall positive remark.
>
> 7. **Assumptions on biological plausibility (concerning computation of global modulator u)** We will add a paragraph on the challenges of how the broad modulation signal could be extracted within the brain. Note that in the field of reinforcement learning, the TD algorithm was around for two decades before the work of Dabney et al. [2020] finally managed to suggest a concrete mapping to biological architecture.
>
> References:
> - J.  Y.  Angela  and  P.  Dayan.   Uncertainty,  neuromodulation,  and  attention.Neuron,  46(4):681–692,2005
> - W.  Dabney,   Z.  Kurth-Nelson,   N.  Uchida,   C.  K.  Starkweather,   D.  Hassabis,   R.  Munos,   andM. Botvinick.  A distributional code for value in dopamine-based reinforcement learning.Nature,577(7792):671–675, 2020.
> - J.  T.  Dudman,  D.  Tsay,  and  S.  A.  Siegelbaum.   A  Role  for  Synaptic  Inputs  at  Distal  Dendrites:Instructive Signals for Hippocampal Long-Term Plasticity.Neuron, 56(5):866–879, dec 2007.  ISSN08966273.  doi:  10.1016/j.neuron.2007.10.020
> - M. Heilbron and F. Meyniel. Confidence resets reveal hierarchical adaptive learning in humans.PLoScomputational biology, 15(4):e1006972, 2019.
> - M. E. Larkum, J. J. Zhu, and B. Sakmann.  A new cellular mechanism for coupling inputs arriving atdifferent cortical layers.Nature, 398(6725):338–341, 1999.  ISSN 00280836.  doi:  10.1038/18686.
> - V. Liakoni, A. Modirshanechi, W. Gerstner, and J. Brea.  Learning in volatile environments with thebayes factor surprise.Neural Computation, 33(2):269–340, 2021
> - G. Major, M. E. Larkum, and J. Schiller. Active properties of neocortical pyramidal neuron dendrites.Annual  review  of  neuroscience,  36:1–24,  jul  2013.   ISSN  1545-4126.   doi:  10.1146/annurev-neuro-062111-150343
> - M. R. Nassar, K. M. Rumsey, R. C. Wilson, K. Parikh, B. Heasly, and J. I. Gold. Rational regulationof learning dynamics by pupil-linked arousal systems.Nature neuroscience, 15(7):1040–1046, 2012.
> - D. Ostwald, B. Spitzer, M. Guggenmos, T. T. Schmidt, S. J. Kiebel, and F. Blankenburg. Evidence forneural encoding of bayesian surprise in human somatosensation.NeuroImage, 62(1):177–188, 2012
> - A. Payeur, J. Guerguiev, F. Zenke, B. A. Richards, and R. Naud. Burst-dependent synaptic plasticitycan coordinate learning in hierarchical circuits.Nature neuroscience, pages 1–10, 2021.
> - R. Urbanczik and W. Senn.  Learning by the dendritic prediction of somatic spiking.Neuron, 81(3):521–528, 2014.
> - P. Schwartenbeck, T. FitzGerald, R. Dolan, and K. Friston.  Exploration, novelty, surprise, and freeenergy minimization.Frontiers in psychology, 4:710, 2013
> - D. L. Yamins and J. J. DiCarlo. Using goal-driven deep learning models to understand sensory cortex.Nature neuroscience, 19(3):356–365, 2016.

---

> > ### Comment · Reviewer_5VGo · 2021-08-25
> > **thank you and an encouragement to add the provided details to the final version**
> >
> > Thank you for the detailed response, particularly for suggesting a mechanism for computing the surprise signal. I believe acknowledging this challenge and discussing potential solutions will greatly improve your manuscript. I don't think anyone sane would expect you to get all the details right at the first try, but it would be a shame not to start the process of getting there.

---

### Official Review · Reviewer_pEVz · 2021-07-23

**Rating:** 7
**Confidence:** 3

**Summary:**

The manuscript proposes a new biologically-inspired learning mechanism for learning deep representations in feedforward circuits, from continuous input streams which in include an active component in which inputs change as a result of known actions (for instance 'saccades'). Exploiting the temporal dependencies in the input, learning can proceed based on local temporal prediction without the need of top-down learning signals (as used in backprop). Abrupt changes in input statistics due to saccades are further exploited for contrastive learning. Because the switch between fixation and saccades are known, one can define a supervised learning objective for the overall optimization. This procedure yields performance that is roughly on par with less biological state of the art methods (GIM), and the paper includes extensive numerical evaluations of its properties and comparisons to other methods.

**Ethical Concerns:**

No concerns.

**Limitations And Societal Impact:**

No concerns.

**Main Review:**

Overall: The emphasis on exploiting temporal input correlations for learning is especially welcomed, for expanding our understanding of biological learning; i like how the combination of prediction and contrastive learning can be incorporated in a single system by using natural input statistics as the main argument. Mechanistically, it's to a large degree the usual suspects (separate dendritic compartment for learning signals, gating the base synapses plasticity, paired with global modulation, etc) but with a new interpretation which is different enough to be able to maybe generate distinguishable experimental predictions.

Detailed comments:
- is there a role for top-down signals that affect lower level representations in these scheme? As i read it, the basic setup is surely feedforward (i get that the learning signals can be top-down, i am asking about feedback signals affecting dynamics -- which admittedly is beyond basic training of a feedforward deep net with an unsupervised objective).
- more generally, is there a way to combine the learning scheme with task specific goals/supervision signals.
- how does the loss work for RNNs exactly? what statistics is it trying to learn in this case? The appendix c seems to provide info on the mechanics but the initial conceptualization is still unclear.
- not sure where the index c in eq 7 comes from
- the planes vs birds example is neat but somewhat cherry picked the structure of other classes looks a lot messier

Originality: This puts together a range of ideas in NN learning in a single learning system; the mechanistic form of the learning dynamics is very similar to previous biologically inspired approximations of backprop (learning signals in dendritic compartment gating plasticity at primary) although differing in some of the details;  the addition of the saccades that directly affect the temporal input correlation structure is novel to the bio-inspired learning literature, to my knowledge, although substantial previous work on contrastive learning exists in the ML literature. The algorithm also seems new in the technical details, e.g. the loss objective formulation.

Significance: What signals drive hierarchical learning in the brain is still a mystery and a focus of intense theoretical research, especially in recent years. The CLAPP procedure adds a potentially novel piece to the puzzle. This may be interesting both for neuroscientists and for machine learning practitioners interested in online/ local learning.

Clarity: The paper is generally well written and easy to read.

**Time Spent Reviewing:**

3

---

> ### Author Response · Authors · 2021-08-09
> **Author response to Reviewer pEVz**
>
> We thank the reviewer for the thorough review and excellent comments. We appreciate the overall positive reaction and the constructive feedback, which we will address in detail below:
>
> 1. **Role of top-down signals** The current setup of our multilayer feedforward network deliberately does not include top-down signals during inference, in line with papers on contrastive predictive coding in the field of machine learning (see beginning of Section 3). However, top-down, as well as recurrent signals are interesting for both neuroscience and AI and a subject of ongoing research. As demonstrated in the paper and discussed below, our approach already extends to Recurrent Neural Networks (RNNs).
>
> 2. **Including supervision** Yes, we see various ways to include supervision/reward into CLAPP. Supervised training of a linear decoder, as done in the paper, is a first step in this direction. A second step is to take the representation learned with CLAPP as a starting point (initialisation) for transfer learning tasks and fine-tune the encoder later with standard BP or reinforcement learning. A third step is to start fine-tuning already during CLAPP training. We will comment on this in the final version.
>
> 3. **Loss for RNNs** The objective that CLAPP optimises, and hence the statistics it learns, is the same for RNNs and feedforward networks: given a context $c^t$, try to predict the future response $z^{t+\delta}$, unless a saccade happened, in which case the prediction should be as different as possible from $z^{t+\delta}$. In our RNNs, the context $c^t$ at time $t$ is computed by a (single) recurrent layer. To avoid the non-local terms of the weight update, that arise specifically with RNNs when back-propagating through time, we use e-prop of Bellec et al. [2020]. We will add a clarification in the final version.
>
> 4. **Index c** As in Van den Oord et al., [2018] the index $c$ stands for *context* layer which predicts the activity of layer $z$.
> In practice, the layer $c$ is either equal to layer $z$ or the layer above ($ z^{l+1} = c^{l}$) depending on the simulation.
> As (too shortly) introduced in l. 172 - 174, the input, summed activation, and incoming weights of the layer $ c$ are denoted respectively $ x^{ c}$, $ a^{ c}$ and $ W^{ c}$.
> Moreover, we also use (and that might have added further confusion) the letter $c_i^t$ to denote the activity of the context neuron $i$ at time $t$.
> Therefore  $\cdot^{c}$ as an upper index refers to the layer whereas  $c$ as a full-size letter refers to neuronal activity.
>
> 5. **Birds vs planes in tSNE** We agree that the discussion of the linear separability between birds and planes in the t-SNE visualization is not representative for the overall quality of the learnt representation.
> We will delete the separation lines from Figure 3.b and remove the corresponding discussion from the main text. In the main text, we will focus on quantitative arguments such as the comparisons in Figure 4 and Table 1 (readout by linear classifier) which is independent from t-SNE.
>
> References:
> - G. Bellec, F. Scherr, A. Subramoney, E. Hajek, D. Salaj, R. Legenstein, and W. Maass.  A solutionto the learning dilemma for recurrent networks of spiking neurons.Nature communications, 11(1):1–15, 2020
> - A. Van den Oord, Y. Li, and O. Vinyals. Representation Learning with Contrastive Predictive Coding.arXiv Prepr., 2018

---

> > ### Comment · Reviewer_pEVz · 2021-08-11
> > **Post-rebuttal feedback.**
> >
> > Thanks for the clarifications.

---

### Decision · Program_Chairs · 2021-09-27

**Decision:**

Accept (Poster)

**Comment:**

This paper proposes a biologically plausible learning algorithm that implements unsupervised (contrastive) learning. The algorithm can learn hierarchical representations. A particular innovation is the use of saccades to generate contrastive samples. Reviewers all agree that the contribution is novel and significant, the manuscript is well-written, and the computational experiments are convincing.